# Empowering GNNs via Edge-Aware Weisfeiler-Leman Algorithm

**Meng Liu**  *mengliu@tamu.edu*
*Department of Computer Science & Engineering*
*Texas A&M University*

**Haiyang Yu**  *haiyang@tamu.edu*
*Department of Computer Science & Engineering*
*Texas A&M University*

**Shuiwang Ji**  *sji@tamu.edu*
*Department of Computer Science & Engineering*
*Texas A&M University*

**Reviewed on OpenReview:** *https://openreview.net/forum?id=VDy6LgErFM*

## Abstract

Message passing graph neural networks (GNNs) are known to have their expressiveness upper-bounded by 1-dimensional Weisfeiler-Leman (1-WL) algorithm. To achieve more powerful GNNs, existing attempts either require *ad hoc* features, or involve operations that incur high time and space complexities. In this work, we propose a *general* and *provably powerful* GNN framework that preserves the *scalability* of the message passing scheme. In particular, we first propose to empower 1-WL for graph isomorphism test by considering edges among neighbors, giving rise to NC-1-WL. The expressiveness of NC-1-WL is shown to be strictly above 1-WL and below 3-WL theoretically. Further, we propose the NC-GNN framework as a differentiable neural version of NC-1-WL. Our simple implementation of NC-GNN is provably as powerful as NC-1-WL. Experiments demonstrate that our NC-GNN performs effectively and efficiently on various benchmarks.

## 1 Introduction

Graph Neural Networks (GNNs) (Gori et al., 2005; Scarselli et al., 2008) have been demonstrated to be effective for various graph tasks. In general, modern GNNs employ a message passing paradigm where the representation of each node is recursively updated by aggregating representations from its neighbors (Atwood & Towsley, 2016; Li et al., 2016; Kipf & Welling, 2017; Hamilton et al., 2017; Veličković et al., 2018; Xu et al., 2019; Gilmer et al., 2017). Such message passing GNNs, however, have been shown to be *at most* as powerful as the 1-dimensional Weisfeiler-Leman (1-WL) algorithm (Weisfeiler & Lehman, 1968) in distinguishing non-isomorphic graphs (Xu et al., 2019; Morris et al., 2019). Thus, message passing GNNs cannot distinguish some simple graphs and detect certain important structural concepts (Chen et al., 2020; Arvind et al., 2020).

The recent efforts to improve the expressiveness of message passing Graph Neural Networks (GNNs) have been focused on high-dimensional WL algorithms (*e.g.*, Morris et al. (2019); Maron et al. (2019)), exploiting subgraph information (*e.g.*, Bodnar et al. (2021a); Zhang & Li (2021)), or adding more distinguishable features (*e.g.*, Murphy et al. (2019); Bouritsas et al. (2022)). As thoroughly discussed in Section 5 and Appendix B, these existing methods either rely on handcrafted/predefined/domain-specific features, or require high computational cost and memory budget. In contrast, this work aims to **develop a *general* GNN framework with *provably expressive* power, while maintaining the *scalability* of the message passing scheme**.

In particular, we first propose an extension of the 1-WL algorithm, namely NC-1-WL, where NC stands for neighbor communication. To be more specific, we incorporate the information of which two neighbors are communicating (*i.e.*, connected) into the graph isomorphism test algorithm. To achieve this, we mathematically model the edges among neighbors as a *multiset of multisets*, in which each edge is represented as a *multiset* of two elements. We show theoretically that the expressiveness of our NC-1-WL in distinguishing non-isomorphic graphs is strictly above 1-WL and below 3-WL. Further, based on NC-1-WL, we introduce the NC-GNN framework, a general and differentiable neural version of NC-1-WL. We provide a simple implementation of NC-GNN that is proved to be as powerful as NC-1-WL. Compared to existing expressive GNNs, our NC-GNN is a general, provably powerful, and, more importantly, scalable framework.

We thoroughly evaluate the performance of our NC-GNN on graph classification and node classification tasks. Our NC-GNN consistently outperforms GIN, which is as powerful as 1-WL, by significant margins on various benchmarks. Remarkably, NC-GNN achieves an impressive absolute margin of 12.0 over GIN in terms of test accuracy on the CLUSTER dataset. Furthermore, NC-GNN performs competitively and often achieves better results, compared to existing expressive GNNs, while being more efficient.

## 2 Preliminaries

We start by introducing notations. We represent an undirected graph as $\mathcal{G} = (V, E, \boldsymbol{X})$, where $V$ is the set of nodes and $E \subseteq V \times V$ denotes the set of edges. We represent an edge $\{v, u\} \in E$ by $(v, u)$ or $(u, v)$ for simplicity. $\boldsymbol{X} = [\boldsymbol{x}_1, \cdots, \boldsymbol{x}_n]^T \in \mathbb{R}^{n \times d}$ is the node feature matrix, where $n = |V|$ is the number of nodes and $\boldsymbol{x}_v \in \mathbb{R}^d$ represents the $d$-dimensional feature of node $v$. $\mathcal{N}_v = \{u \in V | (v, u) \in E\}$ is the set of neighboring nodes of node $v$. A multiset is denoted as $\{\{\cdots\}\}$ and formally defined as follows.

**Definition 1** (Multiset). A *multiset* is a generalized concept of set allowing repeating elements. A multiset $X$ can be formally represented by a 2-tuple as $X = (S_X, m_X)$, where $S_X$ is the underlying set formed by the distinct elements in the multiset and $m_X : S_X \to \mathbb{Z}^+$ gives the *multiplicity* (*i.e.*, the number of occurrences) of the elements. If the elements in the multiset are generally drawn from a set $\mathcal{X}$ (*i.e.*, $S_X \subseteq \mathcal{X}$), then $\mathcal{X}$ is the *universe* of $X$ and we denote it as $X \subseteq \mathcal{X}$ for ease of notation.

**Message passing GNNs**. Modern GNNs usually follow a message passing scheme to learn node representations in graphs (Gilmer et al., 2017). To be specific, the representation of each node is updated iteratively by aggregating the multiset of representations formed by its neighbors. In general, the $\ell$-th layer of a message passing GNN can be expressed as

$$\boldsymbol{a}_v^{(\ell)} = f_{\text{aggregate}}^{(\ell)}\Big(\{\{\boldsymbol{h}_u^{(\ell-1)}|u \in \mathcal{N}_v\}\}\Big), \quad \boldsymbol{h}_v^{(\ell)} = f_{\text{update}}^{(\ell)}\Big(\boldsymbol{h}_v^{(\ell-1)}, \boldsymbol{a}_v^{(\ell)}\Big). \tag{1}$$

$f_{\text{aggregate}}^{(\ell)}$ and $f_{\text{update}}^{(\ell)}$ are the parameterized functions of the $\ell$-th layer. $\boldsymbol{h}_v^{(\ell)}$ is the representation of node $v$ at the $\ell$-th layer and $\boldsymbol{h}_v^{(0)}$ can be initialized as $\boldsymbol{x}_v$. After employing $L$ such layers, the final representation $\boldsymbol{h}_v^{(L)}$ can be used for prediction tasks on each node $v$. For graph-level problems, a graph representation $\boldsymbol{h}_G$ can be obtained by applying a readout function as,

$$\boldsymbol{h}_G = f_{\text{readout}}\Big(\{\{\boldsymbol{h}_v^{(L)}|v \in V\}\}\Big). \tag{2}$$

**Definition 2** (Isomorphism). Two graphs $\mathcal{G} = (V, E, \boldsymbol{X})$ and $\mathcal{H} = (P, F, \boldsymbol{Y})$ are *isomorphic*, denoted as $\mathcal{G} \simeq \mathcal{H}$, if there exists a *bijective* mapping $g : V \to P$ such that $\boldsymbol{x}_v = \boldsymbol{y}_{g(v)}, \forall v \in V$ and $(v, u) \in E$ iff $(g(v), g(u)) \in F$. Graph isomorphism is still an open problem without a known polynomial-time solution.

**Weisfeiler-Leman algorithm**. The Weisfeiler-Leman algorithm (Weisfeiler & Lehman, 1968) provides a hierarchy for the graph isomorphism testing problem. Its 1-dimensional form (*a.k.a.*, 1-WL or color refinement) is a heuristic method that can efficiently distinguish a broad class of non-isomorphic graphs (Babai & Kucera, 1979). 1-WL assigns a color $c_v^{(0)}$ to each node $v$ according to its initial label (or feature)[1] and then iteratively refines the colors until convergence, where the subsets of nodes with the same colors can not be further

---

[1]If there are no initial features or labels, 1-WL assigns the same color to all the nodes in the graph.

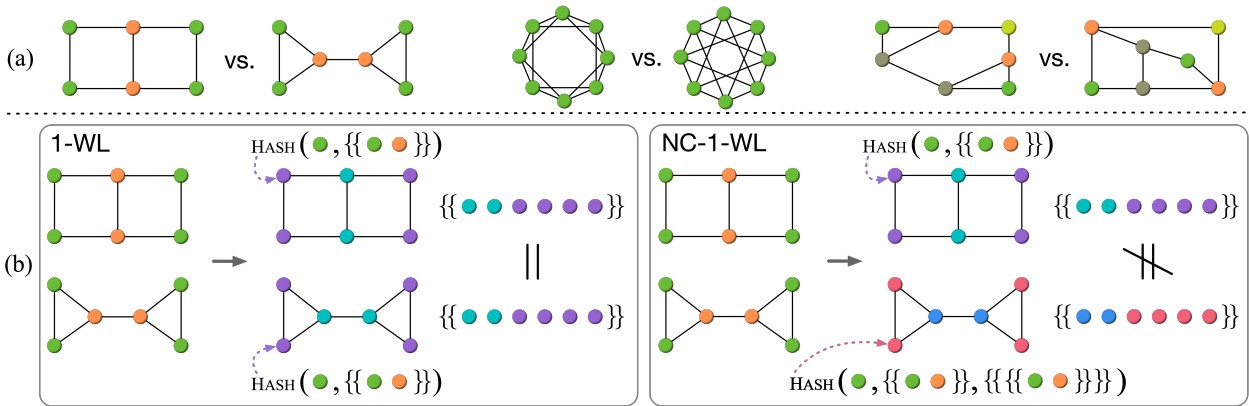

Figure 1: (a) Several example pairs of non-isomorphic graphs, partially adapted from Sato (2020), that cannot be distinguished by 1-WL. Colors represent initial node labels or features. Our NC-1-WL can distinguish them. (b) A comparison between the executions of 1-WL and NC-1-WL on two non-isomorphic graphs.

split to different color groups. In particular, at each iteration $\ell$, it aggregates the colors of nodes and their neighborhoods, which are represented as multisets, and hashes the aggregated results into unique new colors (*i.e.*, injectively). Formally,

$$c_v^{(\ell)} \leftarrow \text{HASH}\Big(c_v^{(\ell-1)}, \{\!\{c_u^{(\ell-1)}|u \in \mathcal{N}_v\}\!\}\Big). \tag{3}$$

1-WL decides two graphs to be non-isomorphic once the colorings between these two graphs differ at some iteration. Instead of coloring each node, $k$-WL generalizes 1-WL by coloring each $k$-tuple of nodes and thus needs to refine the colors for $n^k$ tuples. The details of $k$-WL are provided in Algorithm 2, Appendix A.2. It is known that 1-WL is as powerful as 2-WL in terms of distinguishing non-isomorphic graphs (Cai et al., 1992; Grohe & Otto, 2015; Grohe, 2017). Moreover, for $k \geq 2$, $(k+1)$-WL is strictly more powerful than $k$-WL[2] (Grohe & Otto, 2015). More details of the WL algorithms are given in Cai et al. (1992); Grohe (2017); Sato (2020); Morris et al. (2021).

Given the similarity between message passing GNNs and 1-WL (*i.e.* Eq. (1) *vs.* Eq. (3)), message passing GNNs can be viewed as a differentiable neural version of 1-WL. In fact, the expressiveness of message passing Graph Neural Networks (GNNs) is known to be upper-bounded by 1-WL (Xu et al., 2019; Morris et al., 2019), and message passing GNNs can achieve the same expressiveness as 1-WL if the *aggregate*, *update*, and *readout* functions are injective, which corresponds to the GIN model (Xu et al., 2019). In other words, if two non-isomorphic graphs cannot be distinguished by 1-WL, then message passing GNNs must yield the same embedding for them. Importantly, such expressive power is not sufficient to distinguish some common graphs and cannot capture certain basic structural information such as triangles (Chen et al., 2020; Arvind et al., 2020), which play significant roles in certain tasks, such as tasks over social networks. Several examples that cannot be distinguished by 1-WL or message passing GNNs are shown in Figure 1 (a).

## 3 The NC-1-WL Algorithm

In this section, we propose the NC-1-WL algorithm, which extends the 1-WL algorithm by taking the edges among neighbors into consideration. With such a simple but non-trivial extension, NC-1-WL is proved to be strictly more powerful than 1-WL and less powerful than 3-WL, while preserving the efficiency of 1-WL.

As shown in Eq. (3) and (1), 1-WL and message passing GNNs consider neighbors of each node as a multiset of representations. Here, we move one step forward by further treating edges among neighbors as a multiset,

---

[2]There are two families of WL algorithms; they are $k$-WL and $k$-FWL (Folklore WL). They both consider coloring $k$-tuples and their difference lies in how to aggregate colors from neighboring $k$-tuples. It is known that $(k-1)$-FWL is as powerful as $k$-WL for $k \geq 3$ (Grohe & Otto, 2015; Grohe, 2017; Maron et al., 2019). To avoid ambiguity, in this work, we only involve $k$-WL.

---

**Algorithm 1** NC-1-WL *vs.* 1-WL for graph isomorphism test

---

**Input:** Two graphs $\mathcal{G} = (V, E, \boldsymbol{X})$ and $\mathcal{H} = (P, F, \boldsymbol{Y})$

$c_v^{(0)} \leftarrow \text{HASH}(\boldsymbol{x}_v), \forall v \in V$

$d_p^{(0)} \leftarrow \text{HASH}(\boldsymbol{y}_p), \forall p \in P$

**repeat** $(\ell = 1, 2, \cdots)$

    **if** $\{\!\{c_v^{(\ell-1)} | v \in V\}\!\} \neq \{\!\{d_p^{(\ell-1)} | p \in P\}\!\}$ **then**

        **return** $\mathcal{G} \not\simeq \mathcal{H}$

    **end if**

    **for** $v \in V$ **do**

        $c_v^{(\ell)} \leftarrow \text{HASH}\Big(c_v^{(\ell-1)}, \{\!\{c_u^{(\ell-1)} | u \in \mathcal{N}_v\}\!\}, \underline{\{\!\{\{\!\{c_{u_1}^{(\ell-1)}, c_{u_2}^{(\ell-1)}\}\!\}| u_1, u_2 \in \mathcal{N}_v, (u_1, u_2) \in E\}\!\}}\Big)$

    **end for**

    **for** $p \in P$ **do**

        $d_p^{(\ell)} \leftarrow \text{HASH}\Big(d_p^{(\ell-1)}, \{\!\{d_q^{(\ell-1)} | q \in \mathcal{N}_p\}\!\}, \underline{\{\!\{\{\!\{d_{q_1}^{(\ell-1)}, d_{q_2}^{(\ell-1)}\}\!\}| q_1, q_2 \in \mathcal{N}_p, (q_1, q_2) \in F\}\!\}}\Big)$

    **end for**

**until** convergence

**return** $\mathcal{G} \simeq \mathcal{H}$

---

where each element is also a multiset corresponding to an edge. We formally define a multiset of multisets as follows.

**Definition 3** (Multiset of multisets). A *multiset of multisets*, denoted by $W$, is a multiset where each element is also a multiset. In this work, we only need to consider that each element in $W$ is a multiset formed by 2 elements. Following our definition of multiset, if these 2 elements are generally drawn from a set $\mathcal{X}$, the *universe* of $W$ is the set $\mathcal{W} = \{\{\!\{w_1, w_2\}\!\} | w_1, w_2 \in \mathcal{X}\}$. We can formally represent $W = (S_W, m_W)$, where the underlying set $S_W \subseteq \mathcal{W}$ and $m_W : S_W \to \mathbb{Z}^+$ gives the *multiplicity*. Similarly, we have $W \subseteq \mathcal{W}$.

Particularly, our NC-1-WL considers modeling edges among neighbors as a multiset of multisets and extends 1-WL (*i.e.*, Eq. (3)) to

$$c_v^{(\ell)} \leftarrow \text{HASH}\Big(c_v^{(\ell-1)}, \{\!\{c_u^{(\ell-1)} | u \in \mathcal{N}_v\}\!\}, \underbrace{\boxed{\{\!\{\{\!\{c_{u_1}^{(\ell-1)}, c_{u_2}^{(\ell-1)}\}\!\}| u_1, u_2 \in \mathcal{N}_v, (u_1, u_2) \in E\}\!\}}}_{A \text{ multiset of multisets}}\Big). \quad (4)$$

As 1-WL, our NC-1-WL determines two graphs to be non-isomorphic as long as the colorings of these two graphs are different at some iteration. We summarize the overall process of NC-1-WL in Algorithm 1, where the difference with 1-WL is underlined.

Importantly, our NC-1-WL is more powerful than 1-WL in distinguishing non-isomorphic graphs. Several examples that cannot be distinguished by 1-WL are shown in Figure 1 (a). Our NC-1-WL can distinguish them easily. An example of executions is demonstrated in Figure 1 (b). We rigorously characterize the expressiveness of NC-1-WL by the following theorems. The proofs are given in Appendix A.1 and A.2.

**Theorem 1.** *NC-1-WL is strictly more powerful than 1-WL in distinguishing non-isomorphic graphs.*

**Theorem 2.** *NC-1-WL is strictly less powerful than 3-WL in distinguishing non-isomorphic graphs.*

Although NC-1-WL is less powerful than 3-WL, it is much more efficient. 3-WL has to refine the color of each 3-tuple, resulting in $n^3$ refinement steps in each iteration. In contrast, as 1-WL, NC-1-WL only needs to color each node, which corresponds to $n$ refinement steps in each iteration. Thus, the superiority of our NC-1-WL lies in improving the expressiveness over 1-WL, while being efficient as 1-WL.

Note that our NC-1-WL differs from the concept of Subgraph-1-WL (Zhao et al., 2022), which *ideally* generalizes 1-WL from mapping the neighborhood to mapping the subgraph rooted at each node. Specifically, the refinement step in Subgraph-1-WL is $c_v^{(\ell)} \leftarrow \text{HASH}\big(\mathcal{G}[\mathcal{N}_v^k]\big)$, where $\mathcal{G}[\mathcal{N}_v^k]$ is the subgraph induced by the $k$-hop neighbors of node $v$. However, it requires an injective hash function for subgraphs, which is essentially as hard as the graph isomorphism problem and cannot be achieved. In contrast, our NC-1-WL does not aim

to injectively map the neighborhood subgraph. Instead, we enhance 1-WL by mathematically modeling the edges among neighbors as a multiset of multisets. Then, injectively mapping such multiset of multisets in NC-1-WL is naturally satisfied.

## 4 The NC-GNN Framework

In this section, we propose the NC-GNN framework as a differentiable neural version of NC-1-WL. Further, we establish an instance of NC-GNN that is provably as powerful as NC-1-WL in distinguishing non-isomorphic graphs.

Differing from previous message passing GNNs as Eq. (1), NC-GNN further considers the edges among neighbors as NC-1-WL. One layer of the NC-GNN framework can be formulated as

$$
\begin{aligned}
\boldsymbol{c}_v^{(\ell)} &= f_{\text{communicate}}{}^{(\ell)}\Big(\{\!\{\{\!\{\boldsymbol{h}_{u_1}^{(\ell-1)}, \boldsymbol{h}_{u_2}^{(\ell-1)}\}\!\}| u_1, u_2 \in \mathcal{N}_v, (u_1, u_2) \in E\}\!\}\Big), \\
\boldsymbol{a}_v^{(\ell)} &= f_{\text{aggregate}}{}^{(\ell)}\Big(\{\!\{\boldsymbol{h}_u^{(\ell-1)}| u \in \mathcal{N}_v\}\!\}\Big), \\
\boldsymbol{h}_v^{(\ell)} &= f_{\text{update}}{}^{(\ell)}\Big(\boldsymbol{h}_v^{(\ell-1)}, \boldsymbol{a}_v^{(\ell)}, \boldsymbol{c}_v^{(\ell)}\Big).
\end{aligned}
\tag{5}
$$

$f_{\text{communicate}}{}^{(\ell)}$ is the parameterized function operating on multisets of multisets. The following theorem establishes the conditions under which our NC-GNN can be as powerful as NC-1-WL.

**Theorem 3.** *Let $\mathcal{M} : \mathcal{G} \to \mathbb{R}^d$ be an NC-GNN model with a sufficient number of layers following Eq. (5). $\mathcal{M}$ is as powerful as NC-1-WL in distinguishing non-isomorphic graphs if the following conditions hold: (1) At each layer $\ell$, $f_{communicate}{}^{(\ell)}$, $f_{aggregate}{}^{(\ell)}$, and $f_{update}{}^{(\ell)}$ are injective. (2) The final readout function $f_{readout}$ is injective.*

The proof is provided in Appendix A.3. One may wonder *what advantages NC-GNN has over NC-1-WL*. Note that NC-1-WL only yields different colors to distinguish nodes according to their neighbors and edges among neighbors. These colors, however, do not represent any similarity information and are essentially one-hot encodings. In contrast, NC-GNN, a neural generalization of NC-1-WL, aims at representing nodes in the embedding space. Thus, an NC-GNN model satisfying Theorem 3 can not only distinguish nodes but also learn to map nodes with certain structural similarities to similar embeddings, based on the supervision from the on-hand task. This has the same philosophy as the relationship between message passing GNN and 1-WL (Xu et al., 2019).

There could exist many ways to implement the *communicate*, *aggregate*, and *update* functions in the NC-GNN framework. Here, following the NC-GNN framework, we provide a simple architecture, that provably satisfies Theorem 3 and thus has the same expressive power as NC-1-WL. To achieve this, we generalize the prior results of parameterizing universal functions over *sets* (Zaheer et al., 2017) and *multisets* (Xu et al., 2019) to consider both *multisets* and *multisets of multisets*. Such non-trivial generalization is formalized in the following lemmas. The proofs are available in Appendix A.4 and A.5. As Xu et al. (2019), we assume that the node feature space is countable.

**Lemma 4.** *Assume $\mathcal{X}$ is countable. There exist two functions $f_1$ and $f_2$ so that $h(X, W) = \sum_{x \in X} f_1(x) + \sum_{\{\!\{w_1, w_2\}\!\} \in W} f_2(f_1(w_1) + f_1(w_2))$ is unique for any distinct pair of $(X, W)$, where $X \subseteq \mathcal{X}$ is a multiset with a bounded cardinality and $W \subseteq \mathcal{W} = \{\{\!\{w_1, w_2\}\!\}| w_1, w_2 \in \mathcal{X}\}$ is a multiset of multisets with a bounded cardinality. Moreover, any function $g$ on $(X, W)$ can be decomposed as $g(X, W) = \phi\big(\sum_{x \in X} f_1(x) + \sum_{\{\!\{w_1, w_2\}\!\} \in W} f_2(f_1(w_1) + f_1(w_2))\big)$ for some function $\phi$.*

**Lemma 5.** *Assume $\mathcal{X}$ is countable. There exist two functions $f_1$ and $f_2$ so that for infinitely many choices of $\epsilon$, including all irrational numbers, $h(c, X, W) = (1 + \epsilon)f_1(c) + \sum_{x \in X} f_1(x) + \sum_{\{\!\{w_1, w_2\}\!\} \in W} f_2(f_1(w_1) + f_1(w_2))$ is unique for any distinct 3-tuple of $(c, X, W)$, where $c \in \mathcal{X}$, $X \subseteq \mathcal{X}$ is a multiset with a bounded cardinality, and $W \subseteq \mathcal{W} = \{\{\!\{w_1, w_2\}\!\}| w_1, w_2 \in \mathcal{X}\}$ is a multiset of multisets with a bounded cardinality. Moreover, any function $g$ on $(c, X, W)$ can be decomposed as $g(c, X, W) = \varphi\big((1 + \epsilon)f_1(c) + \sum_{x \in X} f_1(x) + \sum_{\{\!\{w_1, w_2\}\!\} \in W} f_2(f_1(w_1) + f_1(w_2))\big)$ for some function $\varphi$.*

We can use multi-layer perceptrons (MLPs) to model and learn $f_1$, $f_2$, and $\varphi$ in Lemma 5, since MLPs are universal approximators (Hornik et al., 1989; Hornik, 1991). To be specific, we use an MLP to model the compositional function $f_1^{(\ell+1)} \circ \varphi^{(\ell)}$ and another MLP to model $f_2^{(\ell)}$ for $\ell = 1, 2, \cdots, L$. At the first layer, we do not need $f_1^{(1)}$ if the input features are one-hot encodings, since there exists a function $f_2^{(1)}$ that can preserve the injectivity (See Appendix A.5 for details). Overall, one layer of our architecture can be formulated as

$$\boldsymbol{h}_v^{(\ell)} = \text{MLP}_1^{(\ell)}\left(\left(1+\epsilon^{(\ell)}\right)\boldsymbol{h}_v^{(\ell-1)} + \sum_{u \in \mathcal{N}_v} \boldsymbol{h}_u^{(\ell-1)} + \underbrace{\boxed{\sum_{\substack{u_1, u_2 \in \mathcal{N}_v \\ (u_1, u_2) \in E}} \text{MLP}_2^{(\ell)}\left(\boldsymbol{h}_{u_1}^{(\ell-1)} + \boldsymbol{h}_{u_2}^{(\ell-1)}\right)}}_{\textit{The difference with GIN}}\right), \tag{6}$$

where $\epsilon^{(\ell)}$ is a learnable scalar parameter. According to Lemma 5 and Theorem 3, this simple architecture, plus an injective *readout* function, has the same expressive power as NC-1-WL. Note that this architecture follows the GIN model (Xu et al., 2019) closely. The fundamental difference between our model and GIN is highlighted in Eq. (6), which is also our key contribution. Note that if there do not exist any edges among neighbors for all nodes in a graph (*i.e.*, no triangles), the third term in Eq. (6) will be zero for all nodes, and the model will reduce to the GIN model.

**Complexity**. Suppose a graph has $n$ nodes and $m$ edges. Message passing GNNs, such as GIN, require $\mathcal{O}(n)$ memory and have $\mathcal{O}(nd)$ time complexity, where $d$ is the maximum degree of nodes. For each node, we define $\#Message_{NC}$ as the number of edges existing among neighbors of the node. An NC-GNN model as Eq (6) has $\mathcal{O}(n(d+t))$ time complexity, where $t$ denotes the maximum $\#Message_{NC}$ of nodes. Suppose there are totally $T$ triangles in the graph, in addition to $n$ node representations, we need to further store $3T$ representations as the input of $\text{MLP}_2$. If $3T > m$, we can alternatively store $(\boldsymbol{h}_{u_1} + \boldsymbol{h}_{u_2})$ for each edge $(u_1, u_2) \in E$. Thus, the memory complexity is $\mathcal{O}(n + \min(m, 3T))$. Hence, compared to message passing GNNs, our NC-GNN has a bounded memory overhead and preserves the time efficiency. Note that in typical sparse or moderately dense graphs, which are common in many real-world datasets, $t$ is generally small. However, in the worst-case scenario of increasingly dense graphs, it is possible that $t$ is approaching $n^2$, while $d$ is approaching $n$. Thus, in the worst case, NC-GNN has a higher order of time complexity compared to GIN. Despite this, it is crucial to emphasize that even in this worst-case scenario, NC-GNN maintains better efficiency than other high-expressivity GNN models, such as subgraph GNNs. More detailed comparisons are in Section 5 and Appendix B. Overall, compared to many existing expressive GNNs, NC-GNN offers a more favorable balance between expressiveness and scalability.

## 5 Related Work

The most straightforward idea to enhance the expressiveness of message passing GNNs is to mimic the $k$-WL ($k \geq 3$) algorithms (Morris et al., 2019; 2020b; Maron et al., 2019; Chen et al., 2019). For example, Morris et al. (2019) proposes 1-2-3-GNN according to the set-based 3-WL, which is more powerful than 1-WL and less powerful than the tuple-based 3-WL. It requires $\mathcal{O}(n^3)$ memory since representations corresponding to all sets of 3 nodes need to be stored. Moreover, without considering the sparsity of the graph, it has $\mathcal{O}(n^4)$ time complexity since each set aggregates messages from $n$ neighboring sets. Maron et al. (2019) develops PPGN based on high order invariant GNNs (Maron et al., 2018) and 2-FWL, which has the same power as 3-WL. Thereby, PPGN achieves the same power as 3-WL with $\mathcal{O}(n^2)$ memory and $\mathcal{O}(n^3)$ time complexity. Nonetheless, the computational and memory costs of these expressive models are still too high to scale to large graphs.

Another relevant line of research for improving GNNs is to exploit subgraph information (Frasca et al., 2022). Bodnar et al. (2021b;a) perform message passing on high-order substructures of graphs, such as simplicial and cellular complexes. Its preprocessing and message passing step are computationally expensive. GraphSNN (Wijesinghe & Wang, 2022) defines the overlaps between the subgraphs of each node and its neighbors, and then incorporates such overlap information into the message passing scheme by using handcrafted structural coefficients. ESAN (Bevilacqua et al., 2022) employs an equivariant framework to learn from a bag of subgraphs of the graph. Subgraph GNNs Zeng et al. (2021); Sandfelder et al. (2021);

Zhang & Li (2021); Zhao et al. (2022) apply GNNs to the neighborhood subgraph of each node. For example, NGNN (Zhang & Li, 2021) and GNN-AK (Zhao et al., 2022) first apply a base GNN to encode the neighborhood subgraph information of each node and then employ another GNN on the subgraph-encoded representations. Besides, the recently proposed KP-GNN (Feng et al., 2022) focuses on aggregating information from K-hop neighbors of nodes and analyzing its expressive power. It is worth noting that many expressive GNNs, including our proposed NC-GNN, GraphSNN, subgraph GNNs, and KP-GNN, share a common observation that considering more informative structures, *i.e.*, subgraphs, than rooted subtree would enhance the expressive power. Importantly, the key difference lies in how such advanced structures are effectively and efficiently incorporated into deep learning operations. More detailed comparisons to GraphSNN, subgraph GNNs, and KP-GNN are included in Appendix B.

Due to the high memory and time complexity, most of the above methods are usually evaluated on graph-level tasks over small graphs, such as molecular graphs, and can hardly be applied to large graphs like social networks. Compared to these works, our approach differs fundamentally by proposing a *general* (*i.e.*, without *ad hoc* features) and *provably powerful* GNN framework that preserves the *scalability* of regular message passing in terms of computational time and memory requirement. Overall, NC-GNN achieves a sweet spot between expressivity and scalability.

There are several other heuristic methods proposed to strengthen GNNs by adding identity-aware information (Murphy et al., 2019; Vignac et al., 2020; You et al., 2021), random features (Abboud et al., 2021; Dasoulas et al., 2021; Sato et al., 2021), predefined structural features (Monti et al., 2018; Li et al., 2020; Rossi et al., 2020; Bouritsas et al., 2022) to nodes, or randomly drop node (Papp et al., 2021). Another direction is to improve GNNs in terms of the generalization ability (Puny et al., 2020). These works improve GNNs from perspectives orthogonal to ours and thus could be used as techniques to further augment our NC-GNN. In addition, PNA (Corso et al., 2020) applies multiple aggregators to enhance the performance of GNN. Most recently, Morris et al. (2022); Qian et al. (2022); Zhang et al. (2023b); Wang et al. (2023); Zhang et al. (2023a) propose several new hierarchies, which are more fine-grained than the WL hierarchy, for the GNN expressivity problem. For a deeper understanding of expressive GNNs, we recommend referring to the recent surveys (Sato, 2020; Morris et al., 2021; Jegelka, 2022).

## 6 Experiments

In this section, we evaluate the effectiveness of the proposed NC-GNN model on real benchmarks. In particular, we consider widely used datasets from TUDatasets (Morris et al., 2020a), Open Graph Benchmark (OGB) (Hu et al., 2020), and GNN Benchmark (Dwivedi et al., 2020). These datasets are from various domains and cover different tasks over graphs, including graph classification and node classification. Thus, they can provide a comprehensive evaluation of our method. Note that certain datasets in these benchmarks, such as MUTAG, PTC, and NCI1, do not have many edges among neighbors (*i.e.*, Avg. $\#Message_{NC}$ $< 0.2$). In this case, our NC-GNN model will almost reduce to the GIN model and thus perform nearly the same as GIN, as shown in Appendix C.1. Hence, we omit such datasets in our main section. All the used datasets and their statistics, including Avg. $\#Message_{NC}$, are summarized in Table 8, Appendix C.2. Our implementation is based on the PyG library (Fey & Lenssen, 2019). The detailed model configurations and training hyperparameters of NC-GNN on each dataset are summarized in Table 9, Appendix C.3.

**Baselines**. We mainly compare NC-GNN with the following three lines of baselines. (1) **GIN**. As highlighted in Eq. (6), NC-GNN closely follows the GIN model and the fundamental difference is that NC-GNN further considers modeling edges among neighbors. Hence, comparing to GIN can directly demonstrate the effectiveness of including such information in our NC-GNN, which is the core contribution of our theoretical result. We highlight our results if they achieve improvements over GIN. (2) **Subgraph-based GNNs**. As described in Section 5, our NC-GNN and subgraph-based GNNs share the idea of exploiting subgraph information but differ in how to effectively and efficiently incorporate such structures. Thus, we consider two recent subgraph-based GNNs, including GraphSNN (Wijesinghe & Wang, 2022) and GNN-AK (Zhao et al., 2022), into comparison. (3) **Other expressive GNNs**. We further include several other expressive GNNs, including SIN (Bodnar et al., 2021b), CIN (Bodnar et al., 2021a), RingGNN (Chen et al., 2019) and PPGN (Maron et al., 2019). While our goal is to provide a comprehensive comparison, it is challenging to

Table 1: Results (%) on TUDatasets. The top three results on each dataset are highlighted as **first**, **second**, and **third**. We also highlight the cells of NC-GNN results if they are better than GIN.

| Dataset | GIN | GraphSNN | GNN-AK | SIN | CIN | PPGN | NC-GNN |
|---|---|---|---|---|---|---|---|
| COLLAB | **80.2**$_{\pm1.9}$ | - | - | - | - | **81.4**$_{\pm1.4}$ | **82.5**$_{\pm1.2}$ |
| PROTEINS | 76.2$_{\pm2.8}$ | 76.8$_{\pm2.5}$ | **77.1**$_{\pm5.7}$ | 76.5$_{\pm3.4}$ | **77.0**$_{\pm4.3}$ | **77.2**$_{\pm4.7}$ | 76.5$_{\pm4.4}$ |
| IMDB-B | 75.1$_{\pm5.1}$ | **77.9**$_{\pm3.6}$ | 75.0$_{\pm4.2}$ | **75.6**$_{\pm3.2}$ | **75.6**$_{\pm3.7}$ | 73.0$_{\pm5.8}$ | 75.2$_{\pm4.5}$ |
| IMDB-M | 52.3$_{\pm2.8}$ | - | - | **52.5**$_{\pm3.0}$ | **52.7**$_{\pm3.1}$ | 50.5$_{\pm3.6}$ | **52.5**$_{\pm3.2}$ |

include all baselines on every dataset. Additionally, certain baselines may encounter scalability issues when applied to large datasets.

**TUDatasets**. Following GIN (Xu et al., 2019), we conduct experiments on four graph classification datasets where graphs have many edges among neighbors from TUDatasets (Morris et al., 2020a). We use the same number of layers as GIN and report the 10-fold cross-validation accuracy following the protocol as (Xu et al., 2019) for a fair comparison.

As presented in Table 1, we can observe that our NC-GNN outperforms GIN on all datasets consistently. Moreover, NC-GNN performs competitively with other methods that aim to improve the expressiveness of GNN. The consistent improvements over GIN show that modeling edges among neighbors in NC-GNN is practically effective. Notably, NC-GNN achieves an obvious improvement margin of 2.3 on COLLAB. This result can be intuitively justified by observing the considerably larger Avg. $\#Message_{NC}$ on COLLAB compared to other datasets, as shown in Table 8, Appendix C.2. In this case, NC-GNN can use such informative edges existing among neighbors to boost the performance.

**Open Graph Benchmark**. We further perform experiments on the large-scale dataset ogbg-ppa (Hu et al., 2020), which has over 150K graphs and is known as a more convincing testbed than TUDatasets. The graphs in ogbg-ppa are extracted from the protein-protein association networks of different species. Since we have one more MLP than GIN at each layer, one may wonder *if our*

Table 2: Results (%) on ogbg-ppa.

| Model | # Param | Test Acc. |
|---|---|---|
| GIN | 1836942 | 68.92$_{\pm1.00}$ |
| GIN + 3-cycle count feature | - | 70.58$_{\pm0.64}$ |
| GraphSNN | - | 70.66$_{\pm1.65}$ |
| NC-GNN | 1754445 | **71.94**$_{\pm0.43}$ |

*improvements are brought by the larger number of learnable parameters, instead of our claimed expressiveness.* Thus, here we compare with GIN under the same parameter budget. Specifically, we use the same number of layers as GIN to ensure the same receptive field and tune the hidden dimension to obtain an NC-GNN model that has a similar number of learnable parameters as GIN. More detailed setups are included in Appendix C.4. In addition to GIN, we further construct a baseline which is a GIN model with 3-cycle counts included as additional node features. We also include GraphSNN in the comparison. Results over 10 random runs are reported. Note that other subgraph GNN models cannot scale to such a large-scale dataset. For example, NGNN (Zhang & Li, 2021), as pointed out in its paper, does not scale to such a dataset with a large average node degree due to copying a rooted subgraph for each node to the GPU memory.

As reported in Table 2, our NC-GNN model outperforms GIN by an obvious absolute margin of 3.02. Note that the only difference between NC-GNN and GIN is that edges among neighbors are modeled and considered in NC-GNN. Thus, the obvious improvements over GIN can demonstrate that our NC-GNN not only has theoretically provable expressiveness but also achieves good empirical performance on real-world tasks. Furthermore, NC-GNN performs better than using 3-cycle count as features and GraphSNN, which demonstrates that, as detailed in Appendix B.1 and B.2, it is effective in NC-GNN to model edges among neighbors by feature interactions, instead of simply counting 3-cycles as features or using predefined coefficients as GraphSNN.

**GNN Benchmark**. In addition to graph classification tasks, we further experiment with node classification tasks on two datasets, PATTERN and CLUSTER, from GNN Benchmark (Dwivedi et al., 2020). PATTERN and CLUSTER respectively contain 14K and 12K graphs generated from Stochastic Block Models (Abbe, 2017), a widely used mathematical modeling method for studying communities in social networks. The task on these two datasets is to classify nodes in each graph. To be specific, on PATTERN, the goal is to determine

Table 3: Results (%) on GNN Benchmark.

| Model | # Layers | PATTERN | | CLUSTER | |
| --- | --- | --- | --- | --- | --- |
| | | # Param | Test Acc. | # Param | Test Acc. |
| GIN | 4 | 100884 | $85.590_{\pm 0.011}$ | 103544 | $58.384_{\pm 0.236}$ |
| | 16 | 508574 | $85.387_{\pm 0.136}$ | 517570 | $64.716_{\pm 1.553}$ |
| RingGNN | 2 | 105206 | $86.245_{\pm 0.013}$ | 104746 | $42.418_{\pm 20.063}$ |
| | 2 | 504766 | $86.244_{\pm 0.025}$ | 524202 | $22.340_{\pm 0.000}$ |
| | 8 | 505749 | Diverged | 514380 | Diverged |
| PPGN | 3 | 103572 | $85.661_{\pm 0.353}$ | 105552 | $57.130_{\pm 6.539}$ |
| | 3 | 502872 | $85.341_{\pm 0.207}$ | 507252 | $55.489_{\pm 7.863}$ |
| | 8 | 581716 | Diverged | 586788 | Diverged |
| NC-GNN | 4 | 106756 | $\mathbf{86.627}_{\pm 0.017}$ | 107320 | $\mathbf{69.335}_{\pm 0.357}$ |
| | 4 | 506512 | $\mathbf{86.732}_{\pm 0.007}$ | 508428 | $\mathbf{69.838}_{\pm 0.135}$ |
| | 16 | 506512 | $\mathbf{86.607}_{\pm 0.119}$ | 508428 | $\mathbf{76.718}_{\pm 0.071}$ |
| $\Delta^{\uparrow}$ | | | $\mathbf{1.142}\uparrow$ | | $\mathbf{12.002}\uparrow$ |

Table 4: Comparison of real training time per epoch.

| Dataset | PROTEINS | COLLAB | IMDB-B | IMDB-M | PATTERN | CLUSTER |
| --- | --- | --- | --- | --- | --- | --- |
| Avg. $\#Message_{NC}$ | 2.1 | 5016.2 | 59.5 | 70.6 | 3440.1 | 1301.5 |
| GIN | 1.0× | 1.0× | 1.0× | 1.0× | 1.0× | 1.0× |
| PPGN | 8.9× | 1.1× | 4.3× | 4.6× | 19.8× | 23.3× |
| NC-GNN | 1.4× | 1.3× | 1.2× | 1.2× | 6.0× | 4.6× |

if a node belongs to specific predetermined subgraph patterns. On CLUSTER, we aim at categorizing each node to its belonging community. The details of the construction of these datasets are available in (Dwivedi et al., 2020). We compare with GIN, and two methods that mimic 3-WL; those are PPGN and RingGNN. The comparison with subgraph GNNs on these datasets will be presented later in a more comprehensive Table 5. To ensure a fair comparison, we follow Dwivedi et al. (2020) to compare different methods under two budgets of learnable parameters, 100K and 500K, by tuning the number of layers and the hidden dimensions. Average results over 4 random runs are reported in Table 3. On each dataset, we also present the absolute improvement margin of our NC-GNN over GIN, denoted as $\Delta^{\uparrow}$, by comparing their corresponding best result.

We observe that our NC-GNN obtains significant improvements over GIN. To be specific, NC-GNN remarkably outperforms GIN by an absolute margin of 1.142 and 12.002 on PATTERN and CLUSTER, respectively. This further strongly demonstrates the effectiveness of modeling the information of edges among neighbors, which aligns with our theoretical results. Notably, NC-GNN obtains outstanding performance on CLUSTER. Since the task of CLUSTER is to identify communities, we reasonably justify that considering which neighbors are connected is significant for inferring communities. Thus, we believe that our NC-GNN can serve as a robust foundational method for tasks involving social network graphs. Moreover, NC-GNN achieves much better empirical performance than RingGNN and PPGN, although they theoretically mimic 3-WL. It is observed that these 3-WL based methods are difficult to train and thus have fluctuating performance (Dwivedi et al., 2020). In contrast, our NC-GNN is easier to train since it preserves the locality of message passing, thereby being more practically effective.

**Time analysis**. In Table 4, we compare the real training time of models on various datasets, including PROTEINS, COLLAB, IMDB-B, IMDB-M, PATTERN, and CLUSTER. Specifically, we show the increased training time of NC-GNN and PGNN, compared to the training time of GIN. We can observe that our NC-GNN is indeed more efficient than PPGN. Compared to GIN, the increment of the real running time of NC-GNN largely depends on the number of edges among neighbors. For example, the time consumption of NC-GNN is similar to GIN on PROTEINS and IMDB-B, since the Avg. $\#Message_{NC}$ is considerably smaller than that in PATTERN and CLUSTER. Overall, this comparison demonstrates that, compared to GIN, the

Table 5: A thorough comparison between NC-GNN and GIN-AK$^+$ on PATTERN and CLUSTER.

| Datasets | Methods | # Layers | # Param | Test Acc.$^\uparrow$ | Time/epoch$^\downarrow$ | Total Time$^\downarrow$ | GPU Memory$^\downarrow$ | MACS$^\downarrow$ | Inference Time$^\downarrow$ |
|---|---|---|---|---|---|---|---|---|---|
| PATTERN | GIN-AK$^+$ | 4 | 601134 | 86.836$_{\pm 0.007}$ | 77.50s | 0.67h | 31434MB | 176.45G | 32.19s |
| | K-subgraph SAT | 4 | 633586 | 86.845$_{\pm 0.022}$ | 40.73s | 1.66h | 17705MB | 0.03G | 5.91s |
| | NC-GNN | 4 | 552096 | 86.717$_{\pm 0.069}$ | 42.52s | 0.55h | 15625MB | 1.14G | 12.03s |
| CLUSTER | GIN-AK$^+$ | 16 | 602586 | 76.502$_{\pm 0.210}$ | 148.98s | 1.42h | 32142MB | 110.52G | 27.16s |
| | K-subgraph SAT | 16 | 555718 | 77.416$_{\pm 0.269}$ | 74.09s | 3.69h | 21691MB | 0.02G | 4.75s |
| | NC-GNN | 16 | 562948 | 76.992$_{\pm 0.063}$ | 48.87s | 0.68h | 22386MB | 0.84G | 5.18s |

scaling behavior of NC-GNN remains within a reasonable range, particularly when considering the additional expressiveness it provides.

**Thorough comparison with subgraph GNNs and subgraph-aware transformer**. Given that subgraph GNNs are also exploiting neighborhood subgraph information, we further perform a comprehensive empirical comparison with subgraph GNNs, specifically, GIN-AK$^+$ (Zhao et al., 2022), a representative subgraph GNN model. In addition, we include a graph transformer method into comparison, which can position our work within the broader landscape of recent advances in the field. We acknowledge it is hard to compare with various graph transformers (Dwivedi & Bresson, 2020; Kreuzer et al., 2021; Ying et al., 2021; Chen et al., 2022; Hussain et al., 2022; Ma et al., 2023) due to different architectures and various techniques, such as positional encodings. Here, we include the K-subgraph SAT model (Chen et al., 2022) into our comparison, which can utilize a message passing GNN for subgraph encoding before applying global attention modules. This method aligns with our emphasis on exploiting subgraph information and allows us to compare NC-GNN with a state-of-the-art graph transformer method. For each experiment, we run it four times and report the average results for test accuracy, training time per epoch, total time consumed to achieve the best epoch, and GPU memory consumption while keeping the same batch size. In order to compare the computational cost, we use a metric MACS to calculate the average multiply-accumulate operations for each graph in the test set. Lower MACS value typically indicates faster inference. The results are summarized in Table 5. Three models achieve comparable accuracies. Overall, compared to GIN-AK$^+$, NC-GNN is more efficient in training, including training time per epoch and total training time. In addition, we use less GPU memory since we do not have to update node representations for all the nodes in the expanded subgraphs. More importantly, the MACS overhead of GIN-AK$^+$ is 100x more than our NC-GNN. Since NC-GNN calculates each node representation from the original graph instead of the expanded subgraphs, it can save huge MACS overhead during the forward procedure. Thus, NC-GNN takes less time during inference. In comparison with K-subgraph SAT, NC-GNN demonstrates a reduced training time. Note that graph transformer models, like K-subgraph SAT, often require an additional warm-up stage to reach optimal performance levels. This aspect contributes to the overall longer training duration for such models. NC-GNN has similar memory consumption as K-subgraph SAT since there are many edges among neighbors in these two datasets, as shown in Table 8, Appendix C.2. K-subgraph SAT exhibits slightly better efficiency during inference. This could be due to the inherent architectural differences and optimization techniques employed in graph transformer models.

To further compare to subgraph-based GNNs, we experiment on the graph substructure counting task since it is not only an intuitive measurement for expressive power (Chen et al., 2020) but also is quite relevant to practical tasks in biology and social networks. On counting triangles, our NC-GNN outperforms GIN-AK$^+$ with a lower MAE of 0.0081 as opposed to 0.0123. Details are in Appendix C.5.

# 7    Conclusions

In this work, we first present NC-1-WL as a more powerful graph isomorphism test algorithm than 1-WL, by considering the edges among neighbors. Built on our proposed NC-1-WL, we develop NC-GNN, a general, provably powerful, and scalable framework for graph representation learning. In addition to theoretical expressiveness, we empirically demonstrate that NC-GNN achieves outstanding performance on various real benchmarks. Given its simplicity, efficiency, and effectiveness, we anticipate that NC-GNN will become an important base model for learning from graphs, especially social network graphs.

## Acknowledgments

This work was supported in part by National Science Foundation grant IIS-2006861 and National Institutes of Health grant U01AG070112.

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

# A Proofs of Theorems and Lemmas

## A.1 Proof of Theorem 1

**Theorem 1.** *NC-1-WL is strictly more powerful than 1-WL in distinguishing non-isomorphic graphs.*

*Proof.* To prove that NC-1-WL is strictly more powerful than 1-WL, we prove the correctness of the following two statements. (1) If two graphs are determined to be isomorphic by NC-1-WL, then they must be indistinguishable by 1-WL as well. (2) There exist at least two non-isomorphic graphs that cannot be distinguished by 1-WL but can be distinguished by NC-1-WL.

(1) Assume two graphs $\mathcal{G} = (V, E, \boldsymbol{X})$ and $\mathcal{H} = (P, F, \boldsymbol{Y})$ cannot be distinguished by NC-1-WL. Then, according to Algorithm 1, at any iteration $\ell = 1, 2, \cdots$, we have $\left\{\left\{\left(c_v^{(\ell-1)}, \{\{c_u^{(\ell-1)} | u \in \mathcal{N}_v\}\}, \{\{\{\{c_{u_1}^{(\ell-1)}, c_{u_2}^{(\ell-1)}\}\} | u_1, u_2 \in \mathcal{N}_v, (u_1, u_2) \in E\}\}\right) | v \in V\right\}\right\} = \left\{\left\{\left(d_p^{(\ell-1)}, \{\{d_q^{(\ell-1)} | q \in \mathcal{N}_p\}\}, \{\{\{\{d_{q_1}^{(\ell-1)}, d_{q_2}^{(\ell-1)}\}\} | q_1, q_2 \in \mathcal{N}_p, (q_1, q_2) \in F\}\}\right) | p \in P\right\}\right\}$. This naturally implies that, at any iteration $\ell = 1, 2, \cdots$, we have $\left\{\left\{\left(c_v^{(\ell-1)}, \{\{c_u^{(\ell-1)} | u \in \mathcal{N}_v\}\}\right) | v \in V\right\}\right\} = \left\{\left\{\left(d_p^{(\ell-1)}, \{\{d_q^{(\ell-1)} | q \in \mathcal{N}_p\}\}\right) | p \in P\right\}\right\}$. This indicates that 1-WL cannot distinguish graph $\mathcal{G}$ and graph $\mathcal{H}$ as well.

(2) In Figure 1 (a), we provide several pairs as examples to show that there exist such non-isomorphic graphs that can be distinguished by NC-1-WL but cannot be distinguished by 1-WL. $\square$

## A.2 Proof of Theorem 2

**Theorem 2.** *NC-1-WL is strictly less powerful than 3-WL in distinguishing non-isomorphic graphs.*

*Proof.* To prove that NC-1-WL is strictly less powerful than 3-WL, we prove the correctness of the following two statements. (1) If two graphs are determined to be isomorphic by 3-WL, then they must be indistinguishable by NC-1-WL as well. (2) There exist at least two non-isomorphic graphs that cannot be distinguished by NC-1-WL but can be distinguished by 3-WL.

We first describe the details of $k$-WL in Algorithm 2, following Sato (2020). $k$-WL aims at coloring each $k$-tuple of nodes, denoted as $\boldsymbol{v} \in V^k$. The $i$-th neighborhood of each tuple $\boldsymbol{v} = (v_1, v_2, \cdots, v_k)$ is defined as $\mathcal{N}_{\boldsymbol{v},i} = \{(v_1, \cdots, v_{i-1}, s, v_{i+1}, \cdots, v_k) | s \in V\}$. Similarly, The $i$-th neighborhood of each tuple $\boldsymbol{p} = (p_1, p_2, \cdots, p_k)$ is defined as $\mathcal{N}_{\boldsymbol{p},i} = \{(p_1, \cdots, p_{i-1}, t, p_{i+1}, \cdots, p_k) | t \in P\}$. The initial color of each tuple $\boldsymbol{v}$ is determined by the isomorphic type of the subgraph induced by the tuple, *i.e.*, $\mathcal{G}[\boldsymbol{v}]$. (See Sato (2020) for details). Note that the nodes in $\mathcal{G}[\boldsymbol{v}]$ are ordered based on their orders in the tuple $\boldsymbol{v}$. Thus, $\text{HASH}(\mathcal{G}[\boldsymbol{v}]) = \text{HASH}(\mathcal{H}[\boldsymbol{p}])$ iff (a) $\boldsymbol{x}_{v_i} = \boldsymbol{y}_{p_i}$ for $i = 1, 2, \cdots, k$ and (b) $(v_i, v_j) \in E$ iff $(p_i, p_j) \in F$.

(1) Assume two graphs $\mathcal{G} = (V, E, \boldsymbol{X})$ and $\mathcal{H} = (P, F, \boldsymbol{Y})$ are determined to be isomorphic by 3-WL. $\mathcal{G}$ and $\mathcal{H}$ have the same number of nodes[3], denoted as $n$. Then, according to Algorithm 2, we have $\{\{c_{\boldsymbol{v}}^{(0)} | \boldsymbol{v} \in V^k\}\} = \{\{d_{\boldsymbol{p}}^{(0)} | \boldsymbol{p} \in P^k\}\}$. There always exists an injective mapping $g : V \to P$ such that $c_{\boldsymbol{v}}^{(0)} = d_{g(\boldsymbol{v})}^{(0)}$ (*i.e.*, $\text{HASH}(\mathcal{G}[\boldsymbol{v}]) = \text{HASH}(\mathcal{H}[g(\boldsymbol{v})])$), $\forall \boldsymbol{v} \in V^k$. Here, we directly apply $g$ to a tuple $\boldsymbol{v}$ for ease of notation, which means $g(\boldsymbol{v}) = g((v_1, v_2, v_3)) = (g(v_1), g(v_2), g(v_3))$. Without losing generality, we assume $g$ maps $v_j$ to $p_j$ for $j = 1, 2, \cdots, n$. Then, we can obtain the following results.

(a) We can consider the tuples $\boldsymbol{v} = (v_j, v_j, v_j), \forall v_j \in V$. Given $c_{\boldsymbol{v}}^{(0)} = d_{g(\boldsymbol{v})}^{(0)}$, we can derive $\boldsymbol{x}_{v_j} = \boldsymbol{y}_{p_j}, \forall v_j \in V$.

(b) We further consider the tuples $\boldsymbol{v} = (v_j, v_j, v_r), \forall v_j \in V$. According to $c_{\boldsymbol{v}}^{(0)} = d_{g(\boldsymbol{v})}^{(0)}$, we have $\boldsymbol{x}_{v_r} = \boldsymbol{y}_{p_r}$. We can also have $p_r \in \mathcal{N}_{p_j}$ iff $v_r \in \mathcal{N}_{v_j}$ (Otherwise, $c_{\boldsymbol{v}}^{(0)} \neq d_{g(\boldsymbol{v})}^{(0)}$).

---

[3]Two graphs with different numbers of nodes can be easily distinguished by comparing the multisets of node colors, given that the cardinalities of the two multisets are different.

---

**Algorithm 2** $k$-WL for graph isomorphism test

---

**Input:** Two graphs $\mathcal{G} = (V, E, \boldsymbol{X})$ and $\mathcal{H} = (P, F, \boldsymbol{Y})$

$c_{\boldsymbol{v}}^{(0)} \leftarrow \text{HASH}(\mathcal{G}[\boldsymbol{v}]), \forall \boldsymbol{v} \in V^k$

$d_{\boldsymbol{p}}^{(0)} \leftarrow \text{HASH}(\mathcal{H}[\boldsymbol{p}]), \forall \boldsymbol{p} \in P^k$

**repeat** $(\ell = 1, 2, \cdots)$

    **if** $\{\{c_{\boldsymbol{v}}^{(\ell-1)} | \boldsymbol{v} \in V^k\}\} \neq \{\{d_{\boldsymbol{p}}^{(\ell-1)} | \boldsymbol{v} \in P^k\}\}$ **then**

        **return** $\mathcal{G} \not\simeq \mathcal{H}$

    **end if**

    **for** $\boldsymbol{v} \in V^k$ **do**

        $c_{\boldsymbol{v},i}^{(\ell)} = \{\{c_{\boldsymbol{u}}^{(\ell-1)} | \boldsymbol{u} \in \mathcal{N}_{\boldsymbol{v},i}\}\}, \quad \text{for } i = 1, 2, \cdots, k$

        $c_{\boldsymbol{v}}^{(\ell)} \leftarrow \text{HASH}\left(c_{\boldsymbol{v}}^{(\ell-1)}, c_{\boldsymbol{v},1}^{(\ell)}, c_{\boldsymbol{v},2}^{(\ell)}, \cdots, c_{\boldsymbol{v},k}^{(\ell)}\right)$

    **end for**

    **for** $\boldsymbol{p} \in P^k$ **do**

        $d_{\boldsymbol{p},i}^{(\ell)} = \{\{d_{\boldsymbol{q}}^{(\ell-1)} | \boldsymbol{q} \in \mathcal{N}_{\boldsymbol{p},i}\}\}, \quad \text{for } i = 1, 2, \cdots, k$

        $d_{\boldsymbol{p}}^{(\ell)} \leftarrow \text{HASH}\left(d_{\boldsymbol{p}}^{(\ell-1)}, d_{\boldsymbol{p},1}^{(\ell)}, d_{\boldsymbol{p},2}^{(\ell)}, \cdots, d_{\boldsymbol{p},k}^{(\ell)}\right)$

    **end for**

**until** convergence

**return** $\mathcal{G} \simeq \mathcal{H}$

---

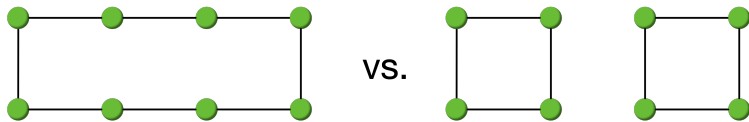

Figure 2: Two graphs, adapted from Sato (2020), that cannot be distinguished by NC-1-WL but can be distinguished by 3-WL.

(c) At last, we consider the tuples $\boldsymbol{v} = (v_j, v_r, v_w), \forall v_j \in V$. Similarly, based on $c_{\boldsymbol{v}}^{(0)} = d_{g(\boldsymbol{v})}^{(0)}$, we can obtain $\boldsymbol{x}_{v_r} = \boldsymbol{y}_{p_r}$ and $\boldsymbol{x}_{v_w} = \boldsymbol{y}_{p_w}$. Also, we can have $p_r, p_w \in \mathcal{N}_{p_j}$ iff $v_r, v_w \in \mathcal{N}_{v_j}$, and $(p_r, p_w) \in F$ iff $(v_r, v_w) \in E$. Now, we consider performing NC-1-WL (Algorithm 1) on these two graphs $\mathcal{G} = (V, E, \boldsymbol{X})$ and $\mathcal{H} = (P, F, \boldsymbol{Y})$ to color each node $v \in V$ and $p \in P$. We have the same injective mapping $g : V \to P$ as above. Based on (a), we have $\boldsymbol{x}_v = \boldsymbol{y}_{g(v)}, \forall v \in V$, which indicates $c_v^{(0)} = d_{g(v)}^{(0)}, \forall v \in V$ in the NC-1-WL coloring process. Similarly, according to (b) and (c), we have $\{\{c_u^{(0)} | u \in \mathcal{N}_v\}\} = \{\{d_q^{(0)} | q \in \mathcal{N}_{g(v)}\}\}, \forall v \in V$ and $\{\{\{c_{u_1}^{(0)}, c_{u_2}^{(0)}\}\} | u_1, u_2 \in \mathcal{N}_v, (u_1, u_2) \in E\}\} = \{\{\{d_{q_1}^{(0)}, d_{q_2}^{(0)}\}\} | q_1, q_2 \in \mathcal{N}_{g(v)}, (q_1, q_2) \in F\}\}, \forall v \in V$, respectively. Therefore, with initial colors satisfying such conditions, NC-1-WL cannot distinguish $\mathcal{G}$ and $\mathcal{H}$ since $c_v^{(l-1)} = d_{g(v)}^{(l-1)}, \forall v \in V$ always holds for $\ell = 1, 2, \cdots$. In other words, $\{\{c_v^{(\ell-1)} | v \in V\}\} = \{\{d_p^{(\ell-1)} | p \in P\}\}$ always holds no matter how many iterations (*i.e.*, $\ell$) we apply.

(2) In Figure 2, we provide two non-isomorphic graphs that can be distinguished by 3-WL but cannot be distinguished by NC-1-WL. For these two graphs, our NC-1-WL reduces to 1-WL since there do not exist any neighbors that are communicating. $\qquad\square$

## A.3 Proof of Theorem 3

**Theorem 3.** *Let $\mathcal{M} : \mathcal{G} \to \mathbb{R}^d$ be an NC-GNN model with a sufficient number of layers following Eq. (5). $\mathcal{M}$ is as powerful as NC-1-WL in distinguishing non-isomorphic graphs if the following conditions hold: (1) At each layer $\ell$, $f_{communicate}^{(\ell)}$, $f_{aggregate}^{(\ell)}$, and $f_{update}^{(\ell)}$ are injective. (2) The final readout function $f_{readout}$ is injective.*

*Proof.* We prove the theorem by showing that an NC-GNN model that satisfies the conditions can yield different embeddings for any two graphs that are determined to be non-isomorphic by NC-1-WL. We denote such model as $\mathcal{M}$. Assume two graphs $\mathcal{G}_1 = (V_1, E_1, \boldsymbol{X}_1)$ and $\mathcal{G}_2 = (V_2, E_2, \boldsymbol{X}_2)$ are determined to be non-isomorphic by NC-1-WL at iteration $L$. Given that $f_{\text{readout}}$ of $\mathcal{M}$ can injectively map different multisets of node features into different embeddings, we only need to demonstrate that $\mathcal{M}$, with a sufficient number of layers, can map $\mathcal{G}_1$ and $\mathcal{G}_2$ to different multisets of node features.

To achieve this, following Xu et al. (2019), we show that, for any iteration $\ell$, there always exists an injective function $\varphi$ such that $\boldsymbol{h}_v^{(\ell)} = \varphi(c_v^{(\ell)})$, where $\boldsymbol{h}_v^{(\ell)}$ is the node representation given by the model $\mathcal{M}$ and $c_v^{(\ell)}$ is the color produced by NC-1-WL. We will show this by induction. Note that here $v$ represents a general node that can be a node in $\mathcal{G}_1$ or $\mathcal{G}_2$.

Let $\phi$ denote the injective hash function used in NC-1-WL. For $\ell = 0$, we have $c_v^{(0)} = \phi(\boldsymbol{x}_v)$ and $\boldsymbol{h}_v^{(0)} = \boldsymbol{x}_v$. Thus, $\varphi$ could be $\phi^{-1}$ for $\ell = 0$. Suppose there exists an injective function $\varphi$ such that $\boldsymbol{h}_v^{(\ell-1)} = \varphi(c_v^{(\ell-1)}), \forall v \in V_1 \cup V_2$, we show that there also exists such an injective function for iteration $\ell$. According to Eq. (5), we have

$$
\begin{aligned}
\boldsymbol{h}_v^{(\ell)} = f_{\text{update}}{}^{(\ell)} &\left( \boldsymbol{h}_v^{(\ell-1)}, f_{\text{aggregate}}{}^{(\ell)} \left( \{\{\boldsymbol{h}_u^{(\ell-1)} | u \in \mathcal{N}_v \}\} \right), \right. \\
&\left. f_{\text{communicate}}{}^{(\ell)} \left( \{\{\{\{\boldsymbol{h}_{u_1}^{(\ell-1)}, \boldsymbol{h}_{u_2}^{(\ell-1)} \}\} | u_1, u_2 \in \mathcal{N}_v, (u_1, u_2) \in E \}\} \right) \right).
\end{aligned}
\tag{7}
$$

According to $\boldsymbol{h}_v^{(\ell-1)} = \varphi(c_v^{(\ell-1)})$, we further have

$$
\begin{aligned}
\boldsymbol{h}_v^{(\ell)} = f_{\text{update}}{}^{(\ell)} &\left( \varphi(c_v^{(\ell-1)}), f_{\text{aggregate}}{}^{(\ell)} \left( \{\{\varphi(c_u^{(\ell-1)}) | u \in \mathcal{N}_v \}\} \right), \right. \\
&\left. f_{\text{communicate}}{}^{(\ell)} \left( \{\{\{\{\varphi(c_{u_1}^{(\ell-1)}), \varphi(c_{u_2}^{(\ell-1)}) \}\} | u_1, u_2 \in \mathcal{N}_v, (u_1, u_2) \in E \}\} \right) \right),
\end{aligned}
\tag{8}
$$

where $f_{\text{communicate}}{}^{(\ell)}$, $f_{\text{aggregate}}{}^{(\ell)}$, $f_{\text{update}}{}^{(\ell)}$, and $\varphi$ are all injective functions. Since the composition of injective functions is also injective, there must exist some injective function $\psi$ such that

$$
\boldsymbol{h}_v^{(\ell)} = \psi \left( c_v^{(\ell-1)}, \{\{c_u^{(\ell-1)} | u \in \mathcal{N}_v \}\}, \{\{\{\{c_{u_1}^{(\ell-1)}, c_{u_2}^{(\ell-1)} \}\} | u_1, u_2 \in \mathcal{N}_v, (u_1, u_2) \in E \}\} \right).
\tag{9}
$$

Then, we can obtain

$$
\begin{aligned}
\boldsymbol{h}_v^{(\ell)} &= \psi \left( \phi^{-1} \left( \phi \left( c_v^{(\ell-1)}, \{\{c_u^{(\ell-1)} | u \in \mathcal{N}_v \}\}, \{\{\{\{c_{u_1}^{(\ell-1)}, c_{u_2}^{(\ell-1)} \}\} | u_1, u_2 \in \mathcal{N}_v, (u_1, u_2) \in E \}\} \right) \right) \right), \\
&= \psi \left( \phi^{-1} \left( c_v^{(\ell)} \right) \right).
\end{aligned}
\tag{10}
$$

Then, $\varphi = \psi \circ \phi^{-1}$ is an injective function such that $\boldsymbol{h}_v^{(\ell)} = \varphi(c_v^{(\ell)}), \forall v \in V_1 \cup V_2$.

Therefore, it is proved that for any iteration $\ell$, there always exists an injective function $\varphi$ such that $\boldsymbol{h}_v^{(\ell)} = \varphi(c_v^{(\ell)})$. Since NC-1-WL determines $\mathcal{G}_1$ and $\mathcal{G}_2$ to be non-isomorphic at iteration $L$, we have $\{\{c_v^{(L)} | v \in V_1 \}\} \neq \{\{c_v^{(L)} | v \in V_2 \}\}$. As proved above, we also have $\{\{\boldsymbol{h}_v^{(L)} | v \in V_1 \}\} = \{\{\varphi(c_v^{(L)}) | v \in V_1 \}\}$, $\{\{\boldsymbol{h}_v^{(L)} | v \in V_2 \}\} = \{\{\varphi(c_v^{(L)}) | v \in V_2 \}\}$, and $\varphi$ is injective. Hence, the multisets of node features produced by $\mathcal{M}$ for $\mathcal{G}_1$ and $\mathcal{G}_2$ are also different, *i.e.*, $\{\{\boldsymbol{h}_v^{(L)} | v \in V_1 \}\} \neq \{\{\boldsymbol{h}_v^{(L)} | v \in V_2 \}\}$, which indicates that the NC-GNN model $\mathcal{M}$ can also distinguish $\mathcal{G}_1$ and $\mathcal{G}_2$. $\qquad\square$

### A.4   Proof of Lemma 4

**Lemma 4.** *Assume $\mathcal{X}$ is countable. There exist two functions $f_1$ and $f_2$ so that $h(X, W) = \sum_{x \in X} f_1(x) + \sum_{\{\{w_1, w_2\}\} \in W} f_2(f_1(w_1) + f_1(w_2))$ is unique for any distinct pair of $(X, W)$, where $X \subseteq \mathcal{X}$ is a multiset with a bounded cardinality and $W \subseteq \mathcal{W} = \{\{\{w_1, w_2\}\} | w_1, w_2 \in \mathcal{X}\}$ is a multiset of multisets with a bounded cardinality. Moreover, any function $g$ on $(X, W)$ can be decomposed as $g(X, W) = \phi\big( \sum_{x \in X} f_1(x) + \sum_{\{\{w_1, w_2\}\} \in W} f_2(f_1(w_1) + f_1(w_2)) \big)$ for some function $\phi$.*

*Proof.* To prove this Lemma, we need the following fact, which is also used by Xu et al. (2019).

**Fact 1**. Assume $\mathcal{X}$ is countable. $h(X) = \sum_{x \in X} N^{-Z(x)}$ is unique for any multiset $X \subseteq \mathcal{X}$ of bounded cardinality, where the mapping $Z : \mathcal{X} \to \mathbb{N}$ is an injection from $x \in \mathcal{X}$ to natural numbers and $N \in \mathbb{N}$ satisfies $N > |X|$ for all $X$.

To prove the correctness of this fact, we show that $X$ can be uniquely obtained from the value of $h(X)$. Following the notations in our main texts, we formally denote $X = (S_X, m_X)$, where $S_X$ is the underlying set of $X$ and $m_X : S_X \to \mathbb{Z}^+$ gives the multiplicity of the elements in $S_X$. Hence, we need to uniquely determine the elements in $S_X$ and their corresponding multiplicities, using the value of $h(X)$. Let $\{x_1, x_2, \cdots, x_n\}$ denote the countable set $\mathcal{X}$ ($n$ could go infinitely). Without losing generality, we assume $Z$ maps $x_1 \to 0$, $x_2 \to 1$, *etc.*. Then we can compute $(q, r) = h(X)$ *divmod* $N^{-0}$, where $q$ is the quotient and $r$ is the remainder. If $q = 0$, we can conclude $x_1$ is not in $S_X$. If $q > 0$, then $x_1$ is in $S_X$ and $q$ gives the multiplicity of $x_1$. Afterwards, we use the remainder $r$ to replace $h(X)$ and compute $(q, r) = h(X)$ *divmod* $N^{-1}$, and the results can be used to infer if $x_2$ is in $S_X$ and its multiplicity. We can do this recursively until $r = 0$. Note that $X$ has a bounded cardinality and $N \in \mathbb{N}$ satisfies $N > |X|$ for all $X$. Otherwise, Fact 1 will not hold. Here we provide an example to show the correctness of Fact 1. Let a multiset $X = \{\{x_1, x_3, x_3\}\}$ and $Z$ injectively maps the elements in $X$ to natural numbers, thus obtaining a multiset $\{\{0, 2, 2\}\}$. Let $N = 4$. We have $h(X) = \sum_{x \in X} N^{-Z(x)} = 4^{-0} + 4^{-2} + 4^{-2} = 9/8$. Next, following our description above, we show how we can infer $X$ by the value of $h(X)$. First, according to $9/8$ *divmod* $4^{-0} = (1, 1/8)$, we can conclude that there is one $x_1$ in $X$. Then, with $1/8$ *divmod* $4^{-1} = (0, 1/8)$, we can infer that $x_2$ is not in $X$. Finally, we have $1/8$ *divmod* $4^{-2} = (2, 0)$, which indicates that there are two $x_3$ in $X$. We can stop the process since the remainder reaches 0.

Let us go back to the proof of Lemma 4. Since $\mathcal{X}$ is countable, $\mathcal{W} = \{\{\{w_1, w_2\}\} | w_1, w_2 \in \mathcal{X}\}$ is also countable. Because both $X$ and $W$ have bounded cardinalities, we can find an number $N \in \mathbb{N}$ such that $N > max(|X| + |W|, 2)$ for all $(X, W)$ pairs. Let $Z_1 : \mathcal{X} \to \mathbb{N}_{odd}$ be an injection from $x \in \mathcal{X}$ to odd natural numbers. We consider $f_1 = N^{-Z_1(x)}$. For ease of notation, we let $\psi(\{\{w_1, w_2\}\}) = f_1(w_1) + f_1(w_2)$. According to Fact 1, $\psi(\{\{w_1, w_2\}\})$ is unique for any $\{\{w_1, w_2\}\} \in \mathcal{W}$. We define the set $\mathcal{Y} = \{\psi(\{\{w_1, w_2\}\}) | w_1, w_2 \in \mathcal{X}\}$. Thus, $\mathcal{Y}$ is also countable as $\mathcal{W}$. We consider $Z_2 : \mathcal{Y} \to \mathbb{N}_{even}$ be an injection from $y \in \mathcal{Y}$ to even natural numbers and $f_2 = N^{-Z_2(y)}$. Then, the resulting $h(X, W) = \sum_{x \in X} f_1(x) + \sum_{\{\{w_1, w_2\}\} \in W} f_2(f_1(w_1) + f_1(w_2))$ is an injective function on $(X, W)$. In other words, we can uniquely determine $(X, W)$ from the value of $h(X, W)$. To be specific, from the value of $h(X, W)$, we can infer the histograms of natural numbers as we show in Fact 1. Then, we can uniquely obtain $X$ (*i.e.*, its underlying set $S_x$ and multiplicities.) based on the odd natural numbers. According to the even natural numbers, we can infer $\{\{\psi(\{\{w_1, w_2\}\}) | \{\{w_1, w_2\}\} \in W\}\}$. Further, since $\psi(\{\{w_1, w_2\}\})$ is injective, we can uniquely obtain $W$.

For any function $g$ on $(X, W)$, we can construct a function $\phi$ such that $\phi\big(h(X, W)\big) = g(X, W)$. This is always achievable since $h(X, W)$ is injective. $\square$

## A.5 Proof of Lemma 5

**Lemma 5.** *Assume $\mathcal{X}$ is countable. There exist two functions $f_1$ and $f_2$ so that for infinitely many choices of $\epsilon$, including all irrational numbers, $h(c, X, W) = (1 + \epsilon)f_1(c) + \sum_{x \in X} f_1(x) + \sum_{\{\{w_1, w_2\}\} \in W} f_2(f_1(w_1) + f_1(w_2))$ is unique for any distinct 3-tuple of $(c, X, W)$, where $c \in \mathcal{X}$, $X \subseteq \mathcal{X}$ is a multiset with a bounded cardinality, and $W \subseteq \mathcal{W} = \{\{\{w_1, w_2\}\} | w_1, w_2 \in \mathcal{X}\}$ is a multiset of multisets with a bounded cardinality. Moreover, any function $g$ on $(c, X, W)$ can be decomposed as $g(c, X, W) = \varphi\big((1 + \epsilon)f_1(c) + \sum_{x \in X} f_1(x) + \sum_{\{\{w_1, w_2\}\} \in W} f_2(f_1(w_1) + f_1(w_2))\big)$ for some function $\varphi$.*

*Proof.* We consider the same functions $f_1 = N^{-Z_1(x)}$ and $f_2 = N^{-Z_2(y)}$ as in our proof for Lemma 4. We prove this lemma by showing that, if $\epsilon$ is an irrational number, $h(c, X, W) \neq h(c', X', W')$ holds for any $(c, X, W) \neq (c', X', W')$. We need to consider the following two cases.

(1) If $c = c'$ but $(X, W) \neq (X', W')$, according to Lemma 4, we have $\sum_{x \in X} f_1(x) + \sum_{\{\{w_1, w_2\}\} \in W} f_2(f_1(w_1) + f_1(w_2)) \neq \sum_{x \in X'} f_1(x) + \sum_{\{\{w_1, w_2\}\} \in W'} f_2(f_1(w_1) + f_1(w_2))$. Thus, we can obtain $h(c, X, W) \neq h(c', X', W')$.

(2) If $c \neq c'$, we show $h(c, X, W) \neq h(c', X', W')$ by contradiction. Assume $h(c, X, W) = h(c', X', W')$, we have

$$
\begin{aligned}
(1 + \epsilon)f_1(c) + \sum_{x \in X} f_1(x) + \sum_{\{\{w_1, w_2\}\} \in W} f_2(f_1(w_1) + f_1(w_2)) = \\
(1 + \epsilon)f_1(c') + \sum_{x \in X'} f_1(x) + \sum_{\{\{w_1, w_2\}\} \in W'} f_2(f_1(w_1) + f_1(w_2)).
\end{aligned} \tag{11}
$$

This can be rewritten as

$$
\begin{aligned}
\epsilon(f_1(c) - f_1(c')) = \Big( f_1(c') + \sum_{x \in X'} f_1(x) + \sum_{\{\{w_1, w_2\}\} \in W'} f_2(f_1(w_1) + f_1(w_2)) \Big) \\
- \Big( f_1(c) + \sum_{x \in X} f_1(x) + \sum_{\{\{w_1, w_2\}\} \in W} f_2(f_1(w_1) + f_1(w_2)) \Big).
\end{aligned} \tag{12}
$$

Since $f_1(c) - f_1(c') \neq 0$ and it is rational, given $\epsilon$ is irrational, we can conclude that L.H.S. of Eq. (12) is irrational. R.H.S. of Eq. (12), however, is rational. This reaches a contradiction. Thus, if $c \neq c'$, we have $h(c, X, W) \neq h(c', X', W')$.

For any function $g$ on $(c, X, W)$, we can construct a function $\varphi$ such that $\varphi\big(h(c, X, W)\big) = g(c, X, W)$. This is always achievable since $h(c, X, W)$ is injective.

**Further justification for the first layer**. If the input features $x \in \mathcal{X}$ are one-hot encodings, $f_1$ is not necessary and thus can be removed. In other words, we can show as follows that there exists an $f_2$ such that $h'(c, X, W) = (1 + \epsilon)c + \sum_{x \in X} x + \sum_{\{\{w_1, w_2\}\} \in W} f_2(w_1 + w_2)$ is unique for any distinct 3-tuple of $(c, X, W)$. Note that $\sum_{x \in X} x$ is injective if input features are one-hot encodings, and the value of $\sum_{x \in X} x$ must be composed of integers. In addition, $\psi'(\{\{w_1, w_2\}\}) = w_1 + w_2$ is also injective. Similarly, We define the set $\mathcal{Y}' = \{\psi'(\{\{w_1, w_2\}\}) | w_1, w_2 \in \mathcal{X}\}$. We consider $Z_2 : \mathcal{Y}' \to \mathbb{N}$ be an injection from $y \in \mathcal{Y}'$ to natural numbers and $f_2 = N^{-Z_2(y)}$, where $N > |W|$ for all $W$. Then $h'(c, X, W) = (1 + \epsilon)c + \sum_{x \in X} x + \sum_{\{\{w_1, w_2\}\} \in W} f_2(w_1 + w_2)$ is injective, since $\sum_{\{\{w_1, w_2\}\} \in W} f_2(w_1 + w_2)$ is unique for any $W$ and is a number $\in (0, 1)$, thus differing from the integer-valued $\sum_{x \in X} x$. This is why we do not need another MLP to model $f_1^{(1)}$ in Eq. (6). $\qquad \square$

# B More Comparisons with Related Work

## B.1 NC-GNN *vs.* Counting 3-Cycles

Since our NC-GNN depends on the existence of 3-cycles in the graph, a natural question is *how our NC-GNN compares to the ability to count 3-cycles*. Here, we theoretically prove that NC-GNN has the ability to count 3-cycles, but not limited (not just equivalent) to that. To achieve this, we show that our NC-GNN (1) has the ability to count 3-cycles and (2) goes beyond just counting 3-cycles (i.e., has the ability to capture patterns that 3-cycles counting cannot). Let's first prove (1). According to the conclusion from Theorem 3 and Lemma 5, the $f_{\text{communicate}}$ in our implemented NC-GNN (Eq. (6)) is injective, thus different $W$ (the multiset of multisets used to denote the edges among neighbors), will lead to distinct outputs. Given that the number of 3-cycles in corresponds to the cardinality of $W$, the 3-cycle counting ability can be included since different cardinalities can be captured by NC-GNN due to the proven injectivity. To prove (2), we can give an example that NC-GNN can differentiate but 3-cycle counting cannot. Assume node $v$ has four neighbors $i, j, p, q$, and two cases $W_1 = \{\{\{\{v_i, v_j\}\}, \{\{v_i, v_p\}\}\}\}$ and $W_2 = \{\{\{\{v_i, v_j\}\}, \{\{v_p, v_q\}\}\}\}$. By solely relying on 3-cycle counting, both scenarios appear identical as both contribute two 3-cycles. In comparison, NC-GNN can differentiate them thanks to the injectivity. Essentially, NC-GNN not only captures the cardinality of $W$ (which is equivalent to counting 3-cycles), but also capture the interaction among neighbors (i.e., the elements in the multiset $W$). More importantly, during training, such interactions are captured via feature interactions in the embedding space and can be learned according to the supervised task at hand.

We empirically verify this in Section 6. As shown in Table 2, NC-GNN exhibits an obvious performance improvement over simple 3-cycle counting.

### B.2  NC-GNN *vs.* GraphSNN

Compared to GraphSNN (Wijesinghe & Wang, 2022), NC-GNN incorporates edges among neighbors in a more general and flexible way, enabling it to capture and model the interactions associated with these edges based on the learning task. To be specific, in GraphSNN, handcrafted structural coefficients are defined to encode overlap subgraphs between neighbors. As defined in Eq. (4) in the GraphSNN paper, the structural coefficient for a specific overlap subgraph is determined by the number of nodes and edges in the overlap subgraph. Such coefficients are handcrafted. Moreover, two different overlap subgraphs with the same number of nodes and edges would have the same structural coefficients, which is less discriminative. In comparison, we mathematically model the edges among neighbors as a multiset of multisets, and such interactions are modeled by feature interactions in the embedding space and can be learned according to the supervised task on hand. Therefore, our model is more general and flexible.

We note that many other expressive GNNs, as well as our NC-GNN, are proved to have expressive power between 1-WL and 3-WL in terms of distinguishing non-isomorphic graphs, but what exact additional power they obtain and how they compare to each other remains unclear and challenging in the community. Having said this, here, we can informally prove that our NC-GNN is more powerful (or not weaker) than GraphSNN.

GraphSNN proposes that overlap-isomorphism lies in between neighborhood subgraph isomorphism and neighborhood subtree isomorphism. However, GraphSNN does not fully solve the local overlap-isomorphism problem. Instead, GraphSNN encodes the overlap subgraph between neighbors with handcrafted structural coefficients. As defined in Eq. (4) in the GraphSNN paper, the structural coefficient for a specific overlap subgraph is determined by the number of nodes and edges in the overlap subgraph. This will lead to the situation that two different overlap subgraphs with the same number of nodes and edges would have the same structural coefficients. In this case, GraphSNN will produce the same structural coefficients. In contrast, our NC-GNN can differentiate these two different overlap subgraphs by capturing the fact that the edges are connecting different nodes in the two overlap subgraphs. This is because of our proved injectivity in Lemma 4 and 5. Because of such injectivity, NC-GNN can tell from the representation which two neighbors are connected (i.e., $f_{communicate}$ in Eq. (5) is injective).

### B.3  NC-GNN *vs.* Subgraph GNNs

Compared to subgraph GNNs such as NGNN (Zhang & Li, 2021) and GNN-AK (Zhao et al., 2022), our NC-GNN is more efficient in terms of time and memory complexity. Subgraph GNNs use a base GNN to encode the neighborhood subgraph of each node and then employ an outer GNN on the subgraph-encoded representations. They need to perform message passing for all nodes in $n$ subgraphs. Note that it is also needed to store many more node representations than regular message passing since the same node in different neighborhood subgraphs has different representations. Thus, the memory complexity is $\mathcal{O}(ns)$ and the time complexity is $\mathcal{O}(nds)$, where $s$ is the maximum number of nodes in the considered $k$-hop neighborhood subgraph. Note that $s$ grows exponentially with the depths of the neighborhood subgraph, thus limiting the scalability. In contrast, our NC-GNN preserves the locality of regular message passing. We still update representations for $n$ nodes in the original graph as regular message passing, instead of all nodes in $n$ subgraphs.

Since the structure considered in NC-GNN for each node form a one-hop subgraph, we further compare our NC-GNN to subgraph GNNs when using the one-hop ego network as the subgraph to encode. Here, we describe the *exact* time and memory consuming (without big $\mathcal{O}$ notations) to explain further why NC-GNN is more efficient than subgraph GNNs in time and memory even when only one-hop subgraphs are used in subgraph GNNs. Let us take NGNN with one-hop ego networks as an example.

**Time**. Let us consider the computational time for obtaining only one node representation. Suppose the node has $d$ neighbors, and there are $t$ edges among the $d$ neighbors. Hence, in the one-hop ego subgraph of this node, there are $(d + t)$ edges in total. Let us only consider the time consumed by the inner GNN module of NGNN and suppose the inner GNN has only one layer of message passing. (This is the simplest NGNN model and the practical model could have more layers of message passing.) Note that the message passing in the inner GNN is performed for all $(d + 1)$ nodes. In other words, each edge in the one-hop ego subgraph

corresponds to two message-passing computations (for the two ending nodes connected by the edge). Hence, the exact time complexity is $2(d + t)$ for NGNN to obtain the representation for this node.

For our NC-GNN, the message passing is only performed for the center node. We aggregate the messages from $d$ neighbors and $t$ edges among neighbors. So the exact time is $(d + t)$. Hence, our NC-GNN only needs half of the time required by (the most efficient version of) NGNN.

**Memory**. For a graph, NC-GNN needs to store $(n + min(m, 3T))$ representations, where $m$ is the number of edges and $T$ is the total number of triangles in the graph. We have a multiplier 3 because each triangle has three edges and their corresponding representations need to be saved for $\text{MLP}_2$ as in Eq. (6). If $3T > m$, we can alternatively save all $(\boldsymbol{h}_v + \boldsymbol{h}_u)$ for any edge $(v, u)$. This is why our additional memory requirement compared to regular message GNN is $min(m, 3T)$.

In comparison, in addition to $n$ node representations, NGNN needs to save $2m$ additional representations. To be specific, for each edge $(v, u)$, it has to save the representation of node $u$ in $v$'s subgraph and the representation of node $v$ in $u$'s subgraph. Hence, NGNN needs to store $(n + 2m)$ representations totally.

Therefore, NGNN need to save $((n + 2m) - (n + min(m, 3T)))$, which is $\geq m$, more representations than our NC-GNN. The number of edges $m$ is usually larger than $n$, which accounts for a large proportion of the total memory cost.

## B.4    NC-GNN *vs.* KP-GNN

NC-GNN and KP-GNN passing share the insight of exploiting subgraph information. However, NC-GNN is distinct from such k-hop message passing (Nikolentzos et al., 2020; Feng et al., 2022). The key differences are following. First, the KP-GNN work mainly focuses on formulating the k-hop message passing framework and analyzing its expressive power. In contrast, we dedicate to the consideration of edges among neighbors, leading to the effective and efficient NC-GNN. Second, the KP-GNN model considers the subgraph information by encoding the number of peripheral edges, while we are modeling the communication among neighbors. In other words, we incorporate feature interaction among neighbors (*i.e.*, the 3rd term in Eq. (6)). With our dedicatedly designed model, our proof of expressivity is totally different from this existing work. Last, in addition to the neural models, we also propose the NC-1-WL as a deterministic algorithm for the graph isomorphism problem, which has not been considered by this existing work.

## B.5    Time Complexity Comparison with Related Work

We compare our NC-GNN with several representative expressive GNNs in Table 6. Our NC-GNN achieves a better balance between expressivity and scalability.

Table 6: Comparison of expressive GNNs. $d$ is the maximum degree of nodes. $T$ is the number of triangles in the graph. $t$ is the maximum $\#Message_{NC}$ of nodes. $s$ is the maximum number of nodes in the neighborhood subgraphs, which grows exponentially with the subgraph depth. It is unknown how the expressiveness upper bound of NGNN compares to 3-WL.

| Method | Memory | Time | Expressiveness upper bound | Scale to large graphs |
|---|---|---|---|---|
| GIN | $\mathcal{O}(n)$ | $\mathcal{O}(nd)$ | 1-WL | ✓ |
| 1-2-3-GNN | $\mathcal{O}(n^3)$ | $\mathcal{O}(n^4)$ | 1-WL $\sim$ 3-WL | - |
| PPGN | $\mathcal{O}(n^2)$ | $\mathcal{O}(n^3)$ | 3-WL | - |
| NGNN | $\mathcal{O}(ns)$ | $\mathcal{O}(nds)$ | 1-WL $\sim$ Unknown | - |
| NC-GNN (ours) | $\mathcal{O}(n + min(m, 3T))$ | $\mathcal{O}(n(d + t))$ | 1-WL $\sim$ 3-WL | ✓ |

Table 7: Results (%) on MUTAG, PTC, and NCI1.

| Dataset | Avg. #Message$_{NC}$ | GIN | NC-GNN |
|---------|---------------------|-----|--------|
| MUTAG | 0 | $89.4_{\pm 5.6}$ | $90.6_{\pm 5.6}$ |
| PTC | 0.005 | $64.6_{\pm 7.0}$ | $64.4_{\pm 5.6}$ |
| NCI1 | 0.005 | $82.7_{\pm 1.7}$ | $82.5_{\pm 1.9}$ |

## C  Experimental Details

### C.1  Comparison on MUTAG, PTC, and NCI1

Graphs in certain datasets do not have many edges among neighbors (*i.e.*, Avg. $\#Message_{NC} < 0.2$). In this case, our NC-GNN model will almost reduce to the GIN model and thus perform nearly the same as GIN. Here, in Table 7, we provide the empirical results on MUTAG, PTC, and NCI1 as examples. As anticipated, the results confirm that NC-GNN performs similarly (with statistical variance) as GIN in scenarios where graphs have sparse connections among neighbors. This is aligned with the theoretical expectations, given that the key innovation of NC-GNN, *i.e.*, the modeling of edges among neighbors, does not have much opportunity to contribute to expressiveness in such graphs.

### C.2  Dataset Statistics

The dataset statistics in shown in Table 8.

Table 8: Dataset Statistics. Avg. #Message$_{NC}$ denotes the average #Message$_{NC}$ per node.

| Dataset | Task | Domain | #Graphs | Avg. #Nodes | Avg. #Edges | Avg. #Message$_{NC}$ |
|---------|------|--------|---------|-------------|-------------|---------------------|
| COLLAB | Graph classification | Social network | 5000 | 74.5 | 4915.6 | 5016.2 |
| PROTEINS | Graph classification | Bioinformatics network | 1113 | 39.1 | 145.6 | 2.1 |
| IMDB-B | Graph classification | Social network | 1000 | 19.7 | 96.5 | 59.5 |
| IMDB-M | Graph classification | Social network | 1500 | 13 | 131.8 | 70.6 |
| ogbg-ppa | Graph classification | Bioinformatics network | 78200/45100/34800 | 243.4 | 2266.1 | 179.3 |
| PATTERN | Node classification | Social network | 10000/2000/2000 | 118.9 | 6079.8 | 3440.1 |
| CLUSTER | Node classification | Social network | 10000/1000/1000 | 117.2 | 4303.8 | 1301.5 |

### C.3  Model Configurations and Training Hyperparameters

For efficiency, we do not tune the model configurations and training hyperparameters for NC-GNN extensively. Since our NC-GNN model is a natural extension of GIN, we usually use the model configurations and tuned hyperparameters of GIN from the community as the starting point for NC-GNN and then perform a grid search for the following hyperparameters according to the validation results.

For the model architecture, we tune the following configurations; those are (1) the number of layers, (2) the number of hidden dimensions, (3) using the jumping knowledge (JK) technique or not, and (4) using a residual connection or not. To ensure a fair comparison, we only consider employing techniques (3) and (4) on the datasets where the baseline GIN model also uses them.

In terms of training, we consider tuning the following hyperparameters. those are (1) the initial learning rate, (2) the step size of learning rate decay, (3) the multiplicative factor of learning rate decay, (4) the batch size, (5) the dropout rate, and (6) the total number of epochs.

The selected model configurations and training hyperparameters for all datasets are summarized in Table 9. For each dataset from GNN Benchmark, we have several NC-GNN models under different parameter budgets, as described in Section 6. Accordingly, we list the configurations and hyperparameters for all of these models for reproducibility.

Table 9: The selected model configurations and training hyperparameters of NC-GNN on all datasets.

| Dataset | COLLAB | PROTEINS | IMDB-B | IMDB-M | ogbg-ppa | PATTERN | CLUSTER |
|---|---|---|---|---|---|---|---|
| # Layers | 5 | 5 | 5 | 5 | 5 | 4/4/16 | 4/4/16 |
| # Hidden Dim. | 64 | 64 | 64 | 32 | 300 | 70/154/78 | 70/154/78 |
| JK | ✓ | ✓ | ✓ | ✓ | - | ✓ | ✓ |
| Residual Con. | - | - | - | - | - | - | - |
| Initial LR | 0.005 | 0.001 | 0.001 | 0.005 | 0.01 | 0.001 | 0.001 |
| Step size of LR | 20 | 50 | 50 | 50 | 30 | 20 | 20 |
| Mul. fac. of LR | 0.5 | 0.5 | 0.5 | 0.5 | 0.1 | 0.5 | 0.5 |
| Batch size | 256 | 32 | 32 | 128 | 32 | 32 | 32 |
| Dropout rate | 0.5 | 0 | 0.5 | 0.5 | 0.5 | 0/0.1/0.1 | 0/0/0.5 |
| # Epochs | 100 | 100 | 200 | 300 | 80 | 100/140/140 | 100/100/200 |

## C.4 Experimental Setup on ogbg-ppa

Differing from TUDatasets, the graphs in ogbg-ppa have edge features representing the type of protein-protein association. In order to apply NC-GNN to these graphs, we further define a variant of our NC-GNN by incorporating edge features into the NC-GNN framework, inspired by the GIN model with edge features introduced by Hu et al. (2019). The layer-wise formulation of our NC-GNN model with considering edge features is

$$\boldsymbol{h}_v^{(\ell)} = \mathrm{MLP}_1^{(\ell)}\Big(\big(1+\epsilon^{(\ell)}\big)\boldsymbol{h}_v^{(\ell-1)} + \sum_{u\in\mathcal{N}_v}\mathrm{ReLU}(\boldsymbol{h}_u^{(\ell-1)}+\boldsymbol{e}_{uv}) + \boxed{\sum_{\substack{u_1,u_2\in\mathcal{N}_v\\(u_1,u_2)\in E}}\mathrm{MLP}_2^{(\ell)}\big(\boldsymbol{h}_{u_1}^{(\ell-1)}+\boldsymbol{h}_{u_2}^{(\ell-1)}+\boldsymbol{e}_{u_1u_2}\big)}\Big),$$

which is a natural extension of Eq. (6) by including edge features. $\boldsymbol{e}_{uv}$ is the edge feature associated with edge $(u,v)$. In practice, we usually apply an embedding layer to input edge features such that they have the same dimension as the node representations.

For reference, the GIN model with considering edge features (Hu et al., 2019) can be formulated as

$$\boldsymbol{h}_v^{(\ell)} = \mathrm{MLP}_1^{(\ell)}\Big(\big(1+\epsilon^{(\ell)}\big)\boldsymbol{h}_v^{(\ell-1)} + \sum_{u\in\mathcal{N}_v}\mathrm{ReLU}(\boldsymbol{h}_u^{(\ell-1)}+\boldsymbol{e}_{uv})\Big).$$

## C.5 Experiments on Counting Triangles

Specifically, following prior work (Chen et al., 2020; Zhao et al., 2022), we use the synthetic dataset from Chen et al. (2020) to evaluate the ability to count triangles in a graph. There are 5000 graphs in total. As (Chen et al., 2020; Zhao et al., 2022), we use the same 30%/20%/50% (training/validation/test) split. We perform 2 random runs and the comparison of average MAE is shown below.

Table 10: Results (%) on counting triangles.

| Model | Test MAE |
|---|---|
| GIN | 0.3569 |
| GIN-AK | 0.0934 |
| GIN-AK$^+$ | 0.0123 |
| NC-GNN | **0.0081**$_{\pm 0.0012}$ |

We compare NC-GNN to GIN and GNN-AK (with GIN as the base model). Our NC-GNN achieves the lowest MAE of 0.0081, outperforming GIN and GIN-AK by a large margin and reducing the MAE by over 90%. We also include the result of GIN-AK$^+$, which incorporates additional strategies such as distance encoding

and context encoding to enhance performance. Despite these improvements, our vanilla NC-GNN model still outperforms GIN-AK$^+$ in counting triangles. According to the results of the above experiment and the results in Table 5, our NC-GNN is more time- and memory-efficient compared to subgraph-based GNNs while achieving practically good performance.

