# OpenReview forum: "Empowering GNNs via Edge-Aware Weisfeiler-Leman Algorithm"
_TMLR — Accepted by TMLR_

### Review · Reviewer_8JPs · 2023-11-14

**Summary Of Contributions:**

This paper introduces an innovative test for determining graph isomorphism. In comparison to the 1-WL test, the proposed test also takes into account the edges connecting the neighbours of the central node in the hash operation. They prove that this new test is strictly more powerful than the 1-WL test without incurring much memory and computational overheads. Motivated by this new algorithm, they devised the corresponding message passing GNNs and empirically showed its superior performance compared to GIN. The paper also highlights that the new algorithm frequently matches or exceeds the test performance of existing more expressive GNNs while benefiting from significantly lower time and memory complexities.

**Audience:**

Yes

**Broader Impact Concerns:**

There aren't any major ethical implications of the work.

**Claims And Evidence:**

Yes

**Requested Changes:**

I am not aware of any critical issues. Authors may refer to the Weaknesses for suggestions to strengthen their work.

**Strengths And Weaknesses:**

## Strengths
1. The paper is well-written and easy to follow.
2. I am not actively tracking the advance in GNNs; however, I believe this is significant work in GNN if the algorithm is new.

## Weaknesses
1. In Section 4 -- Complexity, the time complexity analysis involves $t$, the number of edges existing among neighbours of the nodes. I do not think this parameter is sufficiently natural and authors may discuss its relationship with $d$ and $n$.
2. In Section 6, the performance on datasets with few edges among neighbours is omitted. While the authors mentioned that the performance of NC-GNN will be similar to GIN, it would be more appropriate to include them for completeness and discuss whether we should instead adopt GIN or other architectures in this situation.

---

> ### Author Response · Authors · 2023-12-07
> **Thanks**
>
> Dear reviewer 8JPs, thank you for your constructive feedback. Below are our responses to the raised questions.
>
>
> > In Section 4 -- Complexity, the time complexity analysis involves $t$, the number of edges existing among neighbours of the nodes. I do not think this parameter is sufficiently natural and authors may discuss its relationship with $d$ and $n$.
>
>
> Thank you for the suggestion. We agree that the parameter $t$, the maximum number of edges existing among neighbors of the nodes, requires a more clear discussion concerning its relationship with $d$ (the maximum degree of nodes) and $n$ (the total number of nodes).
>
>
> In typical sparse or moderately dense graphs, which are common in many real-world datasets, $t$ is generally small. In the worst-case scenario of increasingly dense graphs, it is possible that $t$ is approaching $n^2$, while $d$ is approaching $n$. Thus, in the worst case, NC-GNN has a higher order of time complexity compared to GIN. We have included this clarification in the revision.
>
>
> Despite this, it is crucial to emphasize that even in this worst-case scenario, NC-GNN maintains better efficiency than other high-expressivity GNN models, such as subgraph GNNs. As in Appendix B.3, we describe the **exact** time and memory consuming (without big O notations) to explain further NC-GNN is more efficient than subgraph GNNs in time and memory even when only one-hop subgraphs are used in subgraph GNNs.
>
>
> In summary, while NC-GNN does have a higher theoretical complexity in the worst-case scenario compared to GIN, it is important to note that such dense graphs are not typical in the datasets NC-GNN is intended for. Moreover, when it comes to the broader landscape of highly expressive GNNs, NC-GNN offers a more favorable balance between expressiveness and computational efficiency.
>
> > In Section 6, the performance on datasets with few edges among neighbours is omitted. While the authors mentioned that the performance of NC-GNN will be similar to GIN, it would be more appropriate to include them for completeness and discuss whether we should instead adopt GIN or other architectures in this situation.
>
> We appreciate the reviewer’s observation regarding the performance of NC-GNN on datasets with few edges among neighbors, which was not explicitly detailed in Section 6. In response to this feedback, we have included additional performance comparison on such datasets in the revised manuscript. As in Appendix C. 1, we provide the empirical results on MUTAG, PTC, and NCI1 as examples. As anticipated, the results confirm that NC-GNN performs similarly as GIN in scenarios where graphs have sparse connections among neighbors. This is aligned with the theoretical expectations, given that the key innovation of NC-GNN—the modeling of edges among neighbors—does not have much opportunity to contribute to expressiveness in such graphs.
>
> ----
> Thank you for the feedback. Please let us know if there are any further questions.

---

### Review · Reviewer_uBSo · 2023-11-16

**Summary Of Contributions:**

The paper proposes an extension to the Weisfeiler-Leman algorithm (NC-1-WL) that takes into accounts edges between neighbouring nodes. Theory is provided to show that the proposed algorithm has expressiveness between 1-WL and 3-WL. Based on this algorithm, the paper proposes NC-GNN, a graph neural network that is a differentiable neural version, and show that it is as powerful as NC-1-WL. Numerical experiments are provided to show that the proposed approach is effective and efficient.

**Audience:**

Yes

**Broader Impact Concerns:**

Not applicable.

**Claims And Evidence:**

No

**Requested Changes:**

1. [Critical] Please include results that more effectively demonstrate the computational requirements and scaling behaviour of the proposed method. There are strong and important claims in the paper that the proposed method offers a balance between efficiency and power and that it preserves the scalability of message passing. I don't consider that this is supported by the experimental results currently. Even with relatively small graphs, the training time is a factor of 6 greater than GIN. In much larger graphs, where nodes have an average degree of 50, for example, and there is relatively strong clustering behaviour, it seems that the number of edges between neighbours could be high, leading to a relatively dramatic increase in the computational cost.

2. [Critical] Please include some other baseline results that represent the state-of-the-art for the various tasks. This would provide a better indication of how close the proposed method is to achieving state-of-the-art performance. In particular, graph transformers, mixer architectures, or gated networks often tend to be the most effective for the analyzed tasks.

3. [Non-critical] Please include a more in-depth discussion concerning the k-hop GNNs. Currently it is very brief. These would seem to offer similar computational overhead (but I may be wrong on this) and have the same theoretical guarantees. At the moment, there doesn't seem to be a clear experimental comparison and the discussion in the appendix is very brief and vague.

4. [Non-critical] Please expand the discussion on recently proposed alternative hierarchies of expressiveness and, if possible, connect the proposed method to one or more of these hierarchies. In particular the N-WL work seems to me to have connections to the methodology proposed in this paper.

5. [Minor] For reproducibility purposes, it would be helpful to have a more concrete description of the hyperparameter tuning process. It is not possible to reproduce the experimental process of "tune them a little bit according to the validation results."

**Strengths And Weaknesses:**

Strengths

S1. The paper proposes a novel extension of the WL algorithm and show that it is more expressive than 1-WL. In contrast to most of the other similar proposals, the additional computational burden does not appear to be excessive.

S2. Building on this contribution, the paper proposes a novel GNN and shows theoretically that it is as powerful as the proposed WL extension.

S3. The paper reports on numerical experiments for multiple datasets that provide evidence that the proposed technique is efficient and can scale to larger datasets.

S4. The paper is well written and provides a clear description of the proposed method and the experiments that support the claims.

Weaknesses

W1. The paper provides theoretical results demonstrating that the algorithm achieves expressiveness between 1-WL and 3-WL. This is similar to what is achieved by multiple algorithms at this stage. The theoretical results do not provide any further refinement. The proof approaches for Theorems 1 and 2 are relatively standard and straightforward. The proofs of Theorems 3 and Lemmas 4 and 5 are not trivial extensions of the approach in Xu et al. (2019), but they do follow similar ideas. The paper would be considerably stronger if it could provide a richer characterization of the expressiveness achieved by the proposed method. Perhaps connecting it to some of the alternative hierarchies cited in Section 5 would be useful in this regard. For example, is there a connection to the N-WL work of Wang et al. (2023)?

W2. Given that the theoretical characterization of the proposed method does not establish greater expressiveness than multiple other GNNs, the paper relies on empirical results to establish that it achieves similar (or better) performance with less computational overhead.
As a result, we need to rely on the empirical performance to demonstrate that the algorithm offers something beyond existing methods. At one point the paper claims that the proposed method “preserves the scalability of regular message passing in terms of computational time and memory requirement”. The experimental analysis of computation time is limited, with a comparison to only two other algorithms, one being GIN, with experiments still over relatively small datasets. The scaling behaviour is not obvious because already with PATTERN, the proposed algorithm takes approximately 6 times as long as GIN per epoch. The results don't really support the claim that the algorithm "maintains scalability with reasonable overhead" or that it “preserves the scalability of regular message passing”, given that PATTERN and CLUSTER only have an average of approximately 120 nodes per graph.

W3. There is no comparison to other state-of-the-art GNNs such as graph transformers. While it is reasonable to focus on GNNs of a similar class, there should be some comparison with at least some reference GNNs that achieve state-of-the-art performance (including some comparison of computational time and memory overhead). For example, K-Subgraph SAT (Chen et al. 2022 “Structure-Aware Transformer for Graph Representation Learning”), EGT (Hussain et al. 2022, “Self-Attention as a Replacement…”) or GRIT (Ma et al. 2023, “Graph Inductive Biases in Transformers without Message Passing”) offer superior performance for some of the studied datasets. These algorithms may indeed come at a much heavier computational cost or a much larger parameter count, but the paper should provide an indication of this, or a better explanation of why the techniques are not comparable. Some of these papers offer theoretical characterizations, e.g., generalizing WL as in Zhang et al. 2023, as the authors discuss very briefly in Section 5. Given that there is provable expressiveness of some form, it’s not clear why techniques that focus on hierarchies that are “more fine-grained than the WL hierarchy” should be excluded from experimental comparison.

---

> ### Author Response · Authors · 2023-12-07
> **Thanks [Part 2]**
>
> > [Non-critical] Please include a more in-depth discussion concerning the k-hop GNNs.
>
> Thanks for the comment. We made the discussion with k-hop GNNs more extensive in our revised manuscript. Our work and k-hop GNNs do share the insight of exploiting subgraph information. The key differences are summarized below.
>
> (1) The k-hop GNN work mainly focuses on formulating the k-hop message passing framework and analyzing its expressive power. In contrast, we dedicate to the consideration of edges among neighbors, leading to the effective and efficient NC-GNN.
>
> (2) The k-hop GNN model considers the subgraph information by encoding the number of peripheral edges, while we are modeling the communication among neighbors. In other words, we incorporate feature interaction among neighbors (i.e., the 3rd term in our Eq. (6)). With our dedicatedly designed model, our proof of expressivity is totally different from this existing work.
>
> (3) In addition to the neural models, we also propose the NC-1-WL as a deterministic algorithm for the graph isomorphism problem, which has not been considered by k-hop GNN work.
>
> > [Non-critical] Please expand the discussion on recently proposed alternative hierarchies of expressiveness and, if possible, connect the proposed method to one or more of these hierarchies.
>
> Comparing different expressive GNNs theoretically is a complex problem, owing to the coarse granularity of the WL hierarchy. Some recent exciting work [2, 3, 4, 5], as mentioned in Section 5, provide a richer characterization of the expressiveness.
>
> We currently do not have a clear answer and a rigorous proof regarding how NC-GNN compared to these richer hierarchies. We **conjecture** that NC-1-WL and (2,1)-LWL, proposed in [1]’s new hierarchy, have the same expressive power. We provide an intuitive explanation below.
>
> (2,1)-LWL consider coloring all 2-tuples with at most 1 connected component. In other words, it considers tuple $(v, v)$ for all $v \in V$ (1st part) and tuple $(u_1, u_2)$ for each edges $(u_1, u_2)$ (2nd part). The reason why (2,1)-LWL is more powerful than 1-WL is the second part, since the coloring of the 1st part provide the same information as 1-WL. For the coloring of a tuple $(u_1, u_2)$, when we consider replacing $u_2$ with a node connected to $u_2$ (let’s denote the node by $w$ for simplicity), there are only two situations between $u_1$ and $w$; those are, either $u_1$ and $w$ are directly connected in the graph or they are not neighbors. This information is also what we are capturing in NC-1-WL. Therefore, we conjecture that NC-1-WL and (2,1)-LWL has the same expressive power. This is an interesting question and deserves more investigation in the future. While we acknowledge the importance of theoretical comparison, we wish to reiterate the significance of NC-GNN as an efficient and practically powerful base model.
>
>
>
> > [Minor] For reproducibility purposes, it would be helpful to have a more concrete description of the hyperparameter tuning process.
>
> Thank you for the suggestion. We make the hyperparameter tuning process more clear in the revised manuscript.
>
>
>
> ----
> Thank you for the feedback. Please let us know if there are any further questions.
>
> [1] Dexiong Chen, Leslie O’Bray, and Karsten Borgwardt. "Structure-aware transformer for graph representation learning." ICML 2022.
>
> [2] Christopher Morris, Gaurav Rattan, Sandra Kiefer, Siamak Ravanbakhsh. “Speqnets: Sparsity-aware permutation-equivariant graph networks.” ICML 2022.
>
> [3] Chendi Qian, Gaurav Rattan, Floris Geerts, Mathias Niepert, Christopher Morris. “Ordered Subgraph Aggregation Networks.” NeurIPS 2022.
>
> [4] Bohang Zhang, Guhao Feng, Yiheng Du, Di He, Liwei Wang. “A Complete Expressiveness Hierarchy for Subgraph GNNs via Subgraph Weisfeiler-Lehman Tests.” ICML 2023.
>
> [5] Qing Wang, Dillon Ze Chen, Asiri Wijesinghe, Shouheng Li, Muhammad Farhan. “𝒩-WL: A New Hierarchy of Expressivity for Graph Neural Networks.” ICLR 2023.

---

> > ### Comment · Reviewer_uBSo · 2023-12-16
> > **Revised version**
> >
> > Thank you for the thorough response and making the modifications. All of my concerns have been addressed. I think the additional results help to support the claims of the paper and the positioning with regard to other methods. The added commentary should help readers to understand the important contributions of the work.

---

> ### Author Response · Authors · 2023-12-07
> **Thanks [Part 1]**
>
> Dear reviewer uBSo, thank you for your valuable comments. Below are our responses to the raised questions.
>
> > [Critical] Please include results that more effectively demonstrate the computational requirements and scaling behaviour of the proposed method.
>
>
> We recognize the importance of supporting our claims about the efficiency of our proposed NC-GNN model, compared to existing expressive models.
>
> (1) **Comparison with GIN and PPGN.** In Table 4 of the revised manuscript (as included below), we provide a more comprehensive comparison with GIN and PPGN on more datasets in terms of computational efficiency. The revised table shows the increased training time of NC-GNN and PGNN, compared to the training time of GIN.
>
>
>
> |Datasets|PROTEINS|COLLAB|IMDB-B|IMDB-M|PATTERN|CLUSTER|
> |----|----|----|----|----|----|----|
> | Avg. #Message$_\textit{NC}$| $2.1$ | $5016.2$ | $59.5$ | $70.6$ | $3440.1$ | $1301.5$|
> |GIN | $1.0\times$ | $1.0\times$ | $1.0\times$ | $1.0\times$ | $1.0\times$  | $1.0\times$ |
> |PPGN | $8.9\times$ | $1.1\times$ | $4.3\times$ | $4.6\times$ | $19.8\times$ | $23.3\times$|
> |NC-GNN | $1.4\times$ | $1.3\times$ | $1.2\times$ | $1.2\times$ | $6.0\times$  | $4.6\times$  |
>
>
> We can observe that our NC-GNN is indeed more efficient than PPGN. Compared to GIN, the increment of the real running time of NC-GNN largely depends on the number of edges among neighbors. For example, the time consumption of NC-GNN is similar to GIN on PROTEINS and IMDB-B, since the Avg. #Message$_\textit{NC}$ (average number of neighbor-neighbor edges) is considerably smaller than that in PATTERN and CLUSTER. Overall, this comparison demonstrates that, compared to GIN, the scaling behavior of NC-GNN remains within a reasonable range, particularly when considering the additional expressiveness it provides.
>
>
>
> (2) **Comparison with subgraph GNNs.** In Table 5, we offer a thorough comparison between NC-GNN and subgraph GNNs, using multiple metrics to assess efficiency. These results clearly show that NC-GNN is more efficient across these comprehensive metrics, which include training time per epoch, total training time, and GPU memory consumption. In addition, subgraph GNNs cannot scale to the large-scale ogbg-ppa dataset, on which our NC-GNN can be applied. As pointed out in the NGNN paper, it does not scale to such a dataset with a large average node degree due to copying a rooted subgraph for each node to the GPU memory.
>
>
> > [Critical] Please include some other baseline results that represent the state-of-the-art for the various tasks. This would provide a better indication of how close the proposed method is to achieving state-of-the-art performance.
>
>
> Our primary goal was to enhance the expressiveness of GNNs while maintaining computational efficiency. So we initially did not include various graph transformers into comparison. We appreciate the reviewer's suggestion to compare with such methods to position our work within the broader landscape of recent advances in the field.
>
>
> We acknowledge it is hard to compare with various graph transformers due to different architectures and various techniques, such as positional encodings. Here, we include the K-subgraph SAT model [1] into our comparisons, which can utilize a message passing GNN for subgraph encoding before applying global attention modules. This method aligns with our emphasis on exploiting subgraph information and allows us to compare NC-GNN with a state-of-the-art graph transformer method.
>
> Specifically, in the revised manuscript, we expand Table 5 to compare with K-subgraph SAT on the comprehensive metrics. In comparison with K-subgraph SAT, NC-GNN demonstrates a reduced training time. Note that graph transformer models, like K-subgraph SAT, often require an additional warm-up stage to reach optimal performance levels. This aspect contributes to the overall longer training duration for such models. NC-GNN has similar memory consumption as K-subgraph SAT since there are many edges among neighbors in these two datasets, as shown in Table 8, Appendix C.2. K-subgraph SAT exhibits slightly better efficiency during inference. This could be due to the inherent architectural differences and optimization techniques employed in graph transformer models. We believe this expanded analysis will better inform the community about the positioning and potential of NC-GNN within the current landscape of graph representation learning.

---

### Review · Reviewer_15SL · 2023-11-26

**Summary Of Contributions:**

1. This paper proposes a provable and powerful GNN that is overall better than GIN.
2. This paper provides good theoretical support.
3. The experiments are comprehensive to me, including performance and efficiency.

**Audience:**

Yes

**Broader Impact Concerns:**

No such concerns.

**Claims And Evidence:**

Yes

**Requested Changes:**

I am not an expert in graph theory and am not very familiar with prior works in terms of the expressive power of GNN. Here are my suggestions.
1. It seems that the performance/efficiency on graph classification tasks is not so impressive and significant (compared to GIN), at least not as significant as the performance/efficiency on node classification tasks. Could you please give some explanation or discussion?
2. From my understanding, the authors mention that there are GNNs that mimic 3-WL (PPGN, RingGNN) and are more powerful than the proposed methods. It is better to summarize the comparison between these methods and NC-GNN.

**Strengths And Weaknesses:**

1. In Table 1, the results show the proposed method is better than GIN but not so good compared with other methods.

---

> ### Author Response · Authors · 2023-12-07
> **Thanks**
>
> Dear reviewer 15SL, thank you for your constructive review. Below are our responses to the raised questions.
>
> > In Table 1, the results show the proposed method is better than GIN but not so good compared with other methods.
>
> We agree that while NC-GNN shows consistent improvements over GIN, it may not always outperform other methods on these benchmarks. Also, it is worth noting and known in the community that the several datasets in Table 1 exhibit saturation and statistical instability since they are relatively small datasets and obsolete. We included these results for completeness, as they remain a common point of reference in the community. Notably, the improvement on the relatively large dataset in Table 1, i.e., COLLAB, is more statistically significant and can demonstrate modeling edges among neighbors in NC-GNN is practically effective (COLLAB has many edges among neighbors).
>
> Furthermore, our evaluation on the larger, more contemporary datasets like ogbg-ppa, PATTERN, and CLUSTER (Table 2 and 3) showcases the competitive performance of NC-GNN in more complex graph learning tasks, supporting our contributions to the field.
>
> > It seems that the performance/efficiency on graph classification tasks is not so impressive and significant (compared to GIN), at least not as significant as the performance/efficiency on node classification tasks. Could you please give some explanation or discussion?
>
> This is a great question! We think this can be related to two main factors.
>
> **Task Complexity**: Graph classification often involves aggregating node features into a single graph representation. This aggregation step can mask the nuanced differences between nodes that NC-GNN can capture, therefore making it more challenging for the improvements in expressiveness to translate into large performance gains.
>
> **Local Structure Importance**: NC-GNN is particularly good at capturing local neighborhood structures, which is directly beneficial for node classification tasks. These tasks naturally benefit from NC-GNN's ability to distinguish between nodes based on their neighbors and the edges among those neighbors. For example, in the community identification task on CLUSTER, capturing the local neighborhood structures (i.e., which two neighbors are connected) is intuitively effective to infer the community of each node.
>
> Despite this, we maintain that NC-GNN provides an advantage in graph-level tasks, especially in scenarios where local structure is crucial, such as on COLLAB and ogbg-ppa (which are graph classification tasks).
>
> > From my understanding, the authors mention that there are GNNs that mimic 3-WL (PPGN, RingGNN) and are more powerful than the proposed methods. It is better to summarize the comparison between these methods and NC-GNN.
>
>
> Yes, PPGN and RingGNN have a better theoretical expressiveness than NC-GNN since they are mimicking 3-WL. However, it is shown in the benchmark paper [1] that the additional expressiveness of 3-WL mimicking GNNs is hard to be translated into practical performance gains (due to model and training difficulty). As shown in Table 3 and 4, on standard benchmarks, NC-GNN demonstrates superior performance compared to these models, while being more efficient (as shown in our revised Table 4).
>
> ----
> Thank you for the feedback. Please let us know if there are any further questions.
>
> [1] Dwivedi, Vijay Prakash, et al. "Benchmarking Graph Neural Networks." Journal of Machine Learning Research 24.43 (2023): 1-48.

---

> > ### Comment · Reviewer_15SL · 2023-12-10
> >
> > Thank you for your responses. So you mean better expressiveness may not always improve the performance. It makes sense. My major concerns are addressed.

---

### Decision · Action_Editor_1t48 · 2024-01-11

**Recommendation:** Accept as is

**Comment:**

The authors proposed a variant of graph neural networks whose expressive power lies between 1-WL and 3-WL, while claiming similar computational costs as graph message passing. The authors performed thorough theoretical analysis and reasonable experimental comparisons against many existing approaches. The reviewers found the paper well-written, and the authors' response sufficiently addressed most of the initial concerns. While the proposed method does not necessarily have more expressive power than some existing methods, the reviewers nevertheless concluded that the proposed method is novel and the results are of interest to the community. All reviewers recommended acceptance.

One minor comment: Was the full name of NC in NC-1-WL given somewhere?

**Audience:**

Yes, readers who are interested in graph neural networks will find this work relevant and potentially of interest.

**Claims And Evidence:**

The claims are supported by some theoretical analysis and reasonable amount of experimental comparisons.

---

> ### Author Response · Authors · 2024-01-15
>
> Dear Action Editor,
>
> Thank you and the reviewers for the recognition and helpful feedback on our paper. NC means we consider which two `n`eighbors are `c`ommunicating, as mentioned in the 3rd paragraph in the introduction. We will clarify this more explicitly in the camera-ready version. Thank you!
>
> Best,
>
> The authors